# Robust Feature Inference: A Test-time Defense Strategy using Spectral Projections

**Anurag Singh**\*                                                                                   *anurag.singh@cispa.de*
*CISPA Helmholtz Center for Information Security, Saarbrücken, Germany*

**Mahalakshmi Sabanayagam**\*                                                          *sabanaya@cit.tum.de*
*School of Computation, Information and Technology*
*Technical University of Munich, Germany*

**Krikamol Muandet**                                                                             *muandet@cispa.de*
*CISPA Helmholtz Center for Information Security, Saarbrücken, Germany*

**Debarghya Ghoshdastidar**                                                               *ghoshdas@cit.tum.de*
*School of Computation, Information and Technology*
*Technical University of Munich, Germany*

**Reviewed on OpenReview:** *https://openreview.net/forum?id=9OHAtWdFWB*

## Abstract

Test-time defenses are used to improve the robustness of deep neural networks to adversarial examples during inference. However, existing methods either require an additional trained classifier to detect and correct the adversarial samples, or perform additional complex optimization on the model parameters or the input to adapt to the adversarial samples at test-time, resulting in a significant increase in the inference time compared to the base model. In this work, we propose a novel test-time defense strategy called Robust Feature Inference (RFI) that is easy to integrate with any existing (robust) training procedure without additional test-time computation. Based on the notion of robustness of features that we present, the key idea is to project the trained models to the most robust feature space, thereby reducing the vulnerability to adversarial attacks in non-robust directions. We theoretically characterize the subspace of the eigenspectrum of the feature covariance that is the most robust for a generalized additive model. Our extensive experiments on CIFAR-10, CIFAR-100, tiny ImageNet and ImageNet datasets for several robustness benchmarks, including the state-of-the-art methods in RobustBench show that RFI improves robustness across adaptive and transfer attacks consistently. We also compare RFI with adaptive test-time defenses to demonstrate the effectiveness of our proposed approach.

## 1 Introduction

Despite the phenomenal success of deep learning in several challenging tasks, they are prone to vulnerabilities such as the addition of carefully crafted small imperceptible perturbations to the input known as adversarial examples (Szegedy et al., 2013; Goodfellow et al., 2014). While adversarial examples are semantically similar to the input data, they cause the networks to make wrong predictions with high confidence. The primary focus of the community in building adversarially robust models is through modified training procedures. One of the most popular and promising approaches is adversarial training (Madry et al., 2018), which minimizes the maximum loss on the perturbed input samples. Extensive empirical and theoretical studies on the robustness of deep neural networks (DNNs) using adversarial training reveals that adversarial examples are inevitable (Ilyas et al., 2019; Athalye et al., 2018b; Shafahi et al., 2019; Tsipras et al., 2018) and developing

---

*Equal contribution. This work was partly done when AS was a master's student at TU Munich.

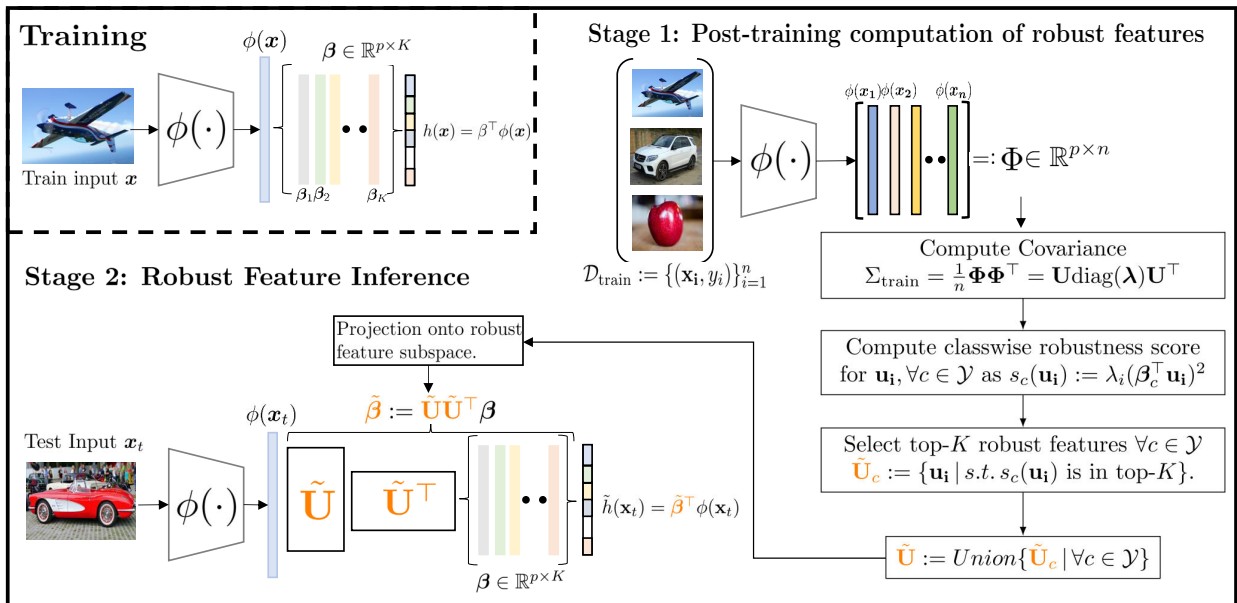

Figure 1: **Illustration of our test-time defense mechanism.** Given any trained model $h(\mathbf{x})$, we first post-process the penultimate layer features $\phi(\mathbf{x})$ to get the top most informative and robust features in eigenspace $\tilde{\mathbf{U}}$ using the training data. During inference of the test data $\mathbf{x}_t$, $\phi(\mathbf{x}_t)$ is projected onto the robust feature space using $\phi(\mathbf{x}_t)\tilde{\mathbf{U}}\tilde{\mathbf{U}}^T$, equivalently changing $\boldsymbol{\beta}$ to $\tilde{\boldsymbol{\beta}} = \tilde{\mathbf{U}}\tilde{\mathbf{U}}^T\boldsymbol{\beta}$.

robust deployable models with safety guarantees require a huge amount of data and model complexity (Nie et al., 2022; Wang et al., 2023; Carmon et al., 2019). For instance, the current state-of-the-art methods (Wang et al., 2023; Peng et al., 2023; Gowal et al., 2021; Rebuffi et al., 2021) use additional one million to 100 million synthetic data along with the original 50000 training samples of CIFAR-10 and CIFAR-100. Although the improvement in robust performance is convincing with this approach, there is an evident tradeoff with huge computational costs both in terms of data and model.

While the deep learning community has focused on achieving robustness through different training paradigms such as adversarial training, little attention has been on improving the robustness of trained models at test-time. *Test-time defenses* refer to methods that improve the robustness of any trained model at test time. This is typically achieved through two main strategies: (i) **Static test-time defenses** update the model parameters or input stochastically independent of the test data (Cohen et al., 2019) or introduce fixed mechanisms to detect and correct the adversarial input (Guo et al., 2018; Nayak et al., 2022). Although the stochastic static defense based on randomized smoothing gives certifiable defense, there is no theoretically well founded deterministic static defense. Most such defenses are based on heuristics. (ii) **Dynamic/Adaptive test-time defenses** adapt the input (Alfarra et al., 2022; Wu et al., 2021) or the model parameters (Kang et al., 2021; Chen et al., 2021) to the test data before making prediction. While the dynamic defenses seem promising as it can adapt to the adversary, the inference is computationally more demanding as it adapts to every single input at test-time. Moreover, the existing adaptive defenses do not necessarily improve the robustness of the underlying model (Croce et al., 2022). Thus, efficiently improving the adversarial robustness of the trained models at test-time without additional data or computation and with theoretical guarantees remain a challenging problem.

**Our contribution.** In this work, we develop a novel test-time defense strategy with the *same inference cost as the underlying model* and no additional data or model complexity. We define robust features, inspired by Athalye et al. (2018b); Ilyas et al. (2019) in Definition 3.1, and subsequently describe the proposed method, RFI, in Algorithm 1 that relies on the idea of retaining the most robust features of the trained model. Notably, RFI is easy to integrate with any existing training paradigm. We provide a theoretical justification for our method by analyzing the robustness of features in a generalized additive model (GAM)

setting (Corollary 3.4). We conduct extensive experiments using different architectures such as ResNet-18, ResNet-50, WideResNet-28-10, WideResNet-34-10, WideResNet-50-2 and PreActResNet-18 on CIFAR-10, CIFAR-100, tiny ImageNet and ImageNet where *RFI yields consistent robustness gains over base models and as well as other adaptive test-time defenses across datasets without additional cost at test time.* Thus, we provide the first theoretically guided method with $1\times$ inference time as the base model, outperforming the adaptive test-time defenses of comparable computation overhead. An interesting by-product of our analysis is the learning dynamics of GAM showing that the features with large variation aligning with the original signal are more robust and learned early during training (Proposition 5.1). This phenomenon has been observed empirically for Neural Tangent Kernel (NTK) features (Tsilivis & Kempe, 2022) without theoretical proof. As a supplementary analysis, we prove it for NTK features (Proposition 5.2).

**Illustration of our method (Figure 1).** The proposed method abstracts any deep neural network as a feature extractor $\phi(\mathbf{x})$ and a linear output layer $\boldsymbol{\beta}^\top \phi(\mathbf{x})$ that consists of class prototypes. We compute the covariance of the features $\Sigma_{\text{train}}$ obtained from the training examples of the feature extractor. We define a robustness measure for the eigenvectors of $\Sigma_{\text{train}}$ as $s_c(\mathbf{u_k})$ and for each class prototype, we retain only the top most eigenvectors with respect to the robustness measure. This choice of the robustness metric as well as the principle of sorting the eigenvector are mathematically justified through Corollary 3.4, where we consider generalized additive models and compute the robustness score of features (Defnition 3.1) showing that the top eigenvectors of the feature matrix are more robust. For a finite-width network, the above theoretical argument essentially corresponds to projecting the weights of only the last layer $\boldsymbol{\beta}$ onto the space spanned by the most robust features $\tilde{\boldsymbol{\beta}} = \tilde{\mathbf{U}}\tilde{\mathbf{U}}^\top \boldsymbol{\beta}$, thereby improving the robustness of the underlying model at test-time. Our method does not increase the inference time because the selected robust eigenbasis can be used to simply transform the linear layer weights into the eigenbasis resulting in exactly the same number of parameters in the network at inference time.

## 2 Related Works

In recent years, there has been a significant amount of research on generating adversarial examples and simultaneously improving the robustness of DNNs against such examples. We review the most relevant works below along with static and adaptive test-time defenses.

**Adversarial robustness.** Szegedy et al. (2013) first observed that the adversarial examples, which are small imperceptible perturbations to the original data, can fool the DNN easily and make incorrect predictions. To generate adversarial examples, Fast Gradient Sign Method (FGSM) is proposed by Goodfellow et al. (2014). Madry et al. (2018) introduced an effective defense against adversarial examples known as adversarial training, where the network is trained by minimizing the maximum loss on the adversarially perturbed inputs. Adversarial training remains a promising defense to significantly improve the robustness of DNNs against adversarial attacks (Rice et al., 2020; Carmon et al., 2019; Engstrom et al., 2019; Wang et al., 2023; Pang et al., 2022). However, sophisticated attacks are developed to break the defenses such as Carlini-Wagner (C&W) attack Carlini & Wagner (2017), a method for generating adversarial examples; 'obfuscated gradients' hypothesis (Athalye et al., 2018b) posits that the vulnerability to adversarial examples is due to the presence of easy to manipulate gradients in the model; 'feature collision' hypothesis Ilyas et al. (2019) postulates that the vulnerability is due to the presence of features in the data that are correlated with the labels, but loses the correlation when perturbed. As an advanced counter defense, methods to constrain the Lipschitzness of the model (Wang et al., 2019) are developed.

**Static test-time defenses.** Static defenses change the model parameters or inputs after training without the knowledge of the test data and remains fixed during inference. A theoretically guaranteed approach to update the model parameters is through randomized smoothing (Cohen et al., 2019; Liu et al., 2018). Another approach is to first detect the adversarial input and correct it using a trained classifier, and input the corrected sample to the base model for prediction. Guo et al. (2018) suggest model agnostic image transformation such as total variance minimization and image quilting for the test data as an effective defense against any adversary. Other works detect the adversarial inputs using a separate trained network and corrects it either by removing the high frequency component in Fourier domain (Nayak et al., 2022) or by a trained masked autoencoder (Chao et al., 2023).

**Adaptive test-time defenses.** Adaptive defenses update model parameters and inputs at inference to defend against the attack. One strategy of adaptive test-time defenses is *input purification*, in which the inputs to a model (usually pre-trained with a robustness objective) is optimized with a test-time objective. This test-time optimization can be hand crafted Alfarra et al. (2022); Wu et al. (2021) or learned Mao et al. (2021); Hwang et al. (2023) with the help of an auxiliary network Nie et al. (2022). Another strategy for building adaptive test-time defenses is *model adaptation*, where model parameters are augmented with activations Chen et al. (2021), implicit representations Kang et al. (2021); Qian et al. (2021) and additional normalization layers Wang et al. (2021). Although several methods are developed for adaptive test-time defenses, all of them increase the inference cost at least 2× (Kang et al., 2021) and sometimes 500× (Shi et al., 2021) compared to the underlying model. More importantly, most of the existing adaptive test-time defenses results in a *weaker adversary than the base model*, hence overestimated the robustness to adaptive attacks and are not really competitive with the static defenses as categorically shown in Croce et al. (2022).

## 3 Robust Feature Inference: A Test-time Defense Strategy using Spectral Projections

We consider multi-class classification problem, where we aim to learn a predictor $h : \mathcal{X} \to \mathcal{Y}$ where $\mathcal{X} \subseteq \mathbb{R}^d$ and $\mathcal{Y} \subset \{0,1\}^C$ is the set of one-hot encodings of $C$ classes. We assume that the data is independent and identically distributed (i.i.d) according to an unknown joint distribution $\mathcal{D}$ over instance-labels $(\mathbf{x}, y) \in \mathcal{X} \times \mathcal{Y}$. The goal of the paper is to develop a test-time defense that can be integrated with any training procedure. Hence, we assume that there exists a learned predictor $h : \mathcal{X} \to \mathcal{Y}$ that we aim to make robust against adversarial attacks. We aim to achieve this by decomposing the predictor into two components, $h(\mathbf{x}) = h_{\mathrm{robust}}(\mathbf{x}) + h_{\mathrm{nonrobust}}(\mathbf{x})$ such that $h_{\mathrm{robust}} : \mathcal{X} \to \mathcal{Y}$ corresponds to the robust component of the predictor while $h_{\mathrm{nonrobust}} : \mathcal{X} \to \mathcal{Y}$ represents the remaining (non robust) component of $h$. In this section, we formally characterize this problem by proposing a notion of robustness of features, inspired by Ilyas et al. (2019), and an algorithm based on pruning less robust features. We also show that the more robust features are more informative.

### 3.1 Robust and Non-Robust Features

The additive decomposition of a predictor in the form $h = h_{\mathrm{robust}} + h_{\mathrm{nonrobust}}$ is difficult in general for predictors with non-linearities in the output, for instance, softmax in multi-class classifiers. Hence, we relax the setup to a multivariate regression problem, that is, $\mathcal{Y} = \mathbb{R}^C$. We further assume that the trained model $h : \mathcal{X} \to \mathcal{Y}$ is given by a generalized additive model (GAM) of the form $h(\mathbf{x}) = \boldsymbol{\beta}^\top \phi(\mathbf{x})$, where $\phi : \mathcal{X} \to \mathcal{H}$ is a smooth function that maps the data into a *feature space* $\mathcal{H}$ and $\boldsymbol{\beta}$ are weights learned in the feature space. The above form of $h$ may represent the solution of kernel regression (with $\mathcal{H}$ being the corresponding reproducing kernel Hilbert space) or $h$ could be the output layer of a neural network, where $\mathcal{H} = \mathbb{R}^p$, $\phi(\mathbf{x})$ denotes the representation learned in the last hidden layer and $\boldsymbol{\beta} \in \mathbb{R}^{p \times C}$ are learned weights of the output layer.

**Features and their robustness.** To identify the robust component of $h$, we aim to approximate $\phi$ as sum of $K$ robust components $(\phi_i)_{i=1}^K$, that is, $h(\mathbf{x}) \approx \sum_{i=1}^K \boldsymbol{\beta}^\top \phi_i(\mathbf{x})$. We refer to each $\phi_i : \mathcal{X} \to \mathcal{H}$ as a *feature*. More generally, we define the set of all features as $\mathcal{F} = \{f : \mathcal{X} \to \mathcal{H}\}$. We now define the robustness of a feature as follows.

**Definition 3.1** ($\ell_2$-Robustness of features)**.** Given a distribution $\mathcal{D}$ on $\mathcal{X} \times \mathbb{R}^C$ and a trained model $h(\mathbf{x}) = \boldsymbol{\beta}^\top \phi(\mathbf{x})$, we define $s_{\mathcal{D}, \boldsymbol{\beta}}(f) = \mathbb{E}_{(\mathbf{x}, y) \sim \mathcal{D}} \left[ \inf_{||\tilde{\mathbf{x}} - \mathbf{x}||_2 \leq \Delta} y^\top \boldsymbol{\beta}^\top f(\tilde{\mathbf{x}}) \right]$ as the robustness of a feature $f \in \mathcal{F}$ and $s_{\mathcal{D}, \boldsymbol{\beta}, c}(f) = \mathbb{E}_{(\mathbf{x}, y) \sim \mathcal{D}} \left[ \inf_{||\tilde{\mathbf{x}} - \mathbf{x}||_2 \leq \Delta} y_c \boldsymbol{\beta}_c^\top f(\tilde{\mathbf{x}}) \right]$ as the robustness of $f$ with respect to the $c$-th class component of $y \in \mathbb{R}^C$, $c \in \{1, \ldots, C\}$, where $\boldsymbol{\beta}_c$ is $c$-th column of $\boldsymbol{\beta}$.

The above definition is based on the notion of robust features introduced by Ilyas et al. (2019) as $\gamma-$robustly useful features, specialized to GAM model. While the $\gamma-$robustly useful feature in Ilyas et al. (2019) is defined on the network output, we define it for the penultimate feature $f$ with a new class-specific definition $s_{\mathcal{D}, \boldsymbol{\beta}, c}(f)$. Based on Definition 3.1, the goal is to approximate $h$ using the most robust features. Searching over all $f \in \mathcal{F}$ is difficult, hence, we focus on features that are linear maps of $\phi$, that is, $f(x) = \boldsymbol{M}^\top \phi(x)$ for

some $\boldsymbol{M} : \mathcal{H} \to \mathcal{H}$ (or $\boldsymbol{M} \in \mathbb{R}^{p \times p}$). For such features, we bound the robustness score from below, under an independent noise model.

**Theorem 3.2** (Lower bound on robustness). *Given $h(\mathbf{x}) = \boldsymbol{\beta}^\top \phi(\mathbf{x})$. Assume that the distribution $\mathcal{D}$ is such that $y = h(\mathbf{x}) + \boldsymbol{\epsilon}$, where $\boldsymbol{\epsilon} \in \mathbb{R}^C$ has independent coordinates, each satisfying $\mathbb{E}[\epsilon_c] = 0$, $\mathbb{E}[\epsilon_c^2] \leq \sigma^2$ for all $c \in \{1, \ldots, C\}$. Further, assume that the map $\phi$ is $L$-Lipschitz, that is, $\|\phi(\mathbf{x}) - \phi(\tilde{\mathbf{x}})\|_{\mathcal{H}} \leq L\|\mathbf{x} - \tilde{\mathbf{x}}\|$. Then, for any $f = \boldsymbol{M}\phi$ and every $c \in \{1, \ldots, C\}$,*

$$s_{\mathcal{D},\boldsymbol{\beta},c}(f) \;\geq\; \boldsymbol{\beta}_c^\top \Sigma \boldsymbol{M} \boldsymbol{\beta}_c - L\Delta \|\boldsymbol{M}\|_{op} \|\boldsymbol{\beta}_c\|_{\mathcal{H}} \sqrt{\sigma^2 + \boldsymbol{\beta}_c^\top \Sigma \boldsymbol{\beta}_c},$$

*where $\Sigma = \mathbb{E}_{\mathbf{x}}\left[\phi(\mathbf{x})\phi(\mathbf{x})^\top\right]$ and $\|\boldsymbol{M}\|_{op}$ is operator norm.*

*Remark 3.3* (Lower bound is tight up to constants). For linear models $\phi(\mathbf{x}) = \mathbf{x}$ and $\mathbb{E}_{\mathbf{x}}[\mathbf{x}] = 0$, $s_{\mathcal{D},\boldsymbol{\beta},c}(f)$ is equal to the lower bound with $L = \frac{2}{\pi}$ (proved in Appendix A.2).

Theorem 3.2 (proved in Appendix A.1) suggests that if we search only over $f \in \mathcal{F}$ that are linear transformations $f = \boldsymbol{M}^\top \phi$ such that $\|\boldsymbol{M}\|_{op} = 1$, then the most robust feature is the one that maximizes the first term $\boldsymbol{\beta}_c^\top \Sigma \boldsymbol{M} \boldsymbol{\beta}_c$. If the search is further restricted to projections onto $K$ dimensional subspace, $\boldsymbol{M} = \boldsymbol{P}\boldsymbol{P}^\top$ with $\boldsymbol{P}$ being the orthonormal basis, then we show that optimizing over such features corresponds to projecting onto the top $K$ eigenvectors $\mathbf{u}$ of $\Sigma$ sorted according to a specific *robustness score*.

**Corollary 3.4.** *Fix any $K$ and $\Sigma = \mathbb{E}_{\mathbf{x}}\left[\phi(\mathbf{x})\phi(\mathbf{x})^\top\right]$. Consider the problem of maximizing the lower bound in Theorem 3.2 over all features $f \in \mathcal{F}$ that correspond to projection of $\phi$ onto $K$ dimensional subspace. Then the solution is $f = \tilde{\mathbf{U}}_c \tilde{\mathbf{U}}_c^\top \phi$ where $\tilde{\mathbf{U}}_c$ is the matrix of the $K$ top eigenvectors of a class-specific matrix $\boldsymbol{B}_c := \frac{1}{2}(\boldsymbol{\beta}_c \boldsymbol{\beta}_c^T \Sigma + \Sigma \boldsymbol{\beta}_c \boldsymbol{\beta}_c^T)$.*

The above result, proved in Appendix A.3, leads to the principle idea of our test-time defense algorithm. The robust output can be defined as $\tilde{h}(\mathbf{x}) = [\boldsymbol{\beta}_1^\top \tilde{\mathbf{U}}_1 \tilde{\mathbf{U}}_1^\top \phi(\mathbf{x}), \ldots, \boldsymbol{\beta}_C^\top \tilde{\mathbf{U}}_C \tilde{\mathbf{U}}_C^\top \phi(\mathbf{x})]$ where $\tilde{\mathbf{U}}_c$ is computed from $\boldsymbol{B}_c$ for every class $c \in \{1, \ldots, C\}$. This results in the most robust projections theoretically, but suffers computationally since it requires $(C + 1)$ eigendecompositions.

**Efficient version.** To improve the computation time, we restrict the search space of $\boldsymbol{M} = \tilde{\mathbf{U}}\tilde{\mathbf{U}}^T$ to the eigenvectors of $\Sigma$, then $\tilde{\mathbf{U}}$ is the matrix of union of $K$ eigenvectors for which the robustness score $s_c(\mathbf{u}_i) = \lambda_i(\boldsymbol{\beta}_c^\top \mathbf{u}_i)^2$ are the largest for every class $c$. This method retains and leverages only the robust features of the trained model at test-time efficiently by projecting the output of the trained model to the eigenspace with higher robustness score. One may naturally ask how much error is incurred by retaining only the robust features. Later, in Corollary 3.6, we discuss that the most robust features also contain most of the information, and hence, drop in performance due to the projection is low.

## 3.2 Our Algorithm: Robust Feature Inference (RFI)

Let $\mathcal{D}_{\text{train}} := \{(\mathbf{x_i}, y_i)\}_{i=1}^n \subset \mathcal{X} \times \mathcal{Y}$ be a training dataset with $n$ samples, and $h : \mathcal{X} \to \mathcal{Y}$ a trained model such that $h(\mathbf{x}) = \boldsymbol{\beta}^\top \phi(\mathbf{x})$ for all $\mathbf{x} \in \mathcal{X}$ where $\phi : \mathcal{X} \to \mathbb{R}^p$ is the feature map defined by the hidden layers of the model $h$ and $\boldsymbol{\beta} \in \mathbb{R}^{p \times C}$ is the weight matrix defined by the last fully-connected layer with $p$ as the dimension of the feature space (refer Figure 1). From the previous analysis, we propose a method operating on the feature space of $\phi$ that projects the features in the robust directions, hence improving robustness by reducing the chance of attacks using the non-robust feature directions. To this end, we first compute the corresponding covariance matrix $\Sigma_{\text{train}}$ of the hidden-layer features based on the input data from $\mathcal{D}_{\text{train}}$, that is,

$$\Sigma_{\text{train}} = \frac{1}{n} \Phi \Phi^\top \text{ with } \Phi := [\phi(\mathbf{x_1}), \ldots, \phi(\mathbf{x_n})] \in \mathbb{R}^{p \times n}.$$

Next, as presented in Figure 1, we compute the eigendecomposition of the covariance $\Sigma_{\text{train}} = \mathbf{U}\text{diag}(\boldsymbol{\lambda})\mathbf{U}^\top$ where $\mathbf{U} \in \mathbb{R}^{p \times p}$ is the matrix whose columns consist of eigenvectors of $\Sigma_{\text{train}}$ denoted by $\mathbf{u_i} \in \mathbb{R}^p$, $\boldsymbol{\lambda} \in \mathbb{R}^p$ is a vector of corresponding eigenvalues such that $\lambda_1 \geq \lambda_2 \geq \ldots$, and $\text{diag}(\boldsymbol{\lambda})$ is a diagonal matrix with eigenvalues as its diagonal entries. The idea of our algorithm is to retain only robustly useful features, i.e., top $K$ eigenvectors, when making predictions on unseen data. For each class $c \in \mathcal{Y}$, we define the $c$-th column of $\boldsymbol{\beta}$ as $\boldsymbol{\beta}_c$ as class prototype for $c \in \mathcal{Y}$. The classwise robustness score of each feature is computed according

---

**Algorithm 1** Robust Feature Inference (RFI)

---

**Require:** The model $h$ trained on $\mathcal{D}_{\text{train}} := \{(\mathbf{x_i}, y_i)\}_{i=1}^n$ such that $h(\mathbf{x}) = \boldsymbol{\beta}^\top \phi(\mathbf{x})$ where $\phi : \mathcal{X} \to \mathbb{R}^p$ and $\boldsymbol{\beta} \in \mathbb{R}^{p \times C}$, and the number of top robust features to select $K$.

1: Compute the covariance $\Sigma_{\text{train}} \leftarrow \frac{1}{n} \Phi \Phi^\top$ where $\Phi := [\phi(x_1), \dots, \phi(x_n)] \in \mathbb{R}^{p \times n}$.

2: Compute eigendecomposition of $\Sigma_{\text{train}} = \mathbf{U} \text{diag}(\boldsymbol{\lambda}) \mathbf{U}^\top$ where columns of $\mathbf{U}$ are $\mathbf{u_i} \in \mathbb{R}^p$. $\{\triangleright$ Top $K$ most robust features $\tilde{\mathbf{U}} \in \mathbb{R}^{p \times K}$ $\qquad (3 \to 7)\}$

3: $\tilde{\mathbf{U}} \leftarrow \{\}$, $\boldsymbol{\beta}_c \leftarrow c$-th column of $\boldsymbol{\beta}$

4: **for** $c \leftarrow 1$ to $C$ **do**

5:     For all $i$, compute robustness score $s_c(\mathbf{u_i}) \leftarrow \lambda_i (\boldsymbol{\beta}_c^T \mathbf{u_i})^2$

6:     $\tilde{\mathbf{U}} \leftarrow \text{Union}(\tilde{\mathbf{U}}, \mathbf{u_i})$ if $s_c(\mathbf{u_i})$ is in top $K$ scores $s_c(.)$

7: **end for**

    $\{\triangleright$ Robust Feature Inference on test set $X_{test}$ $\qquad\qquad\qquad (8 \to 9)\}$

8: $\tilde{\boldsymbol{\beta}} = \tilde{\mathbf{U}} \tilde{\mathbf{U}}^T \boldsymbol{\beta}$

9: $\forall \mathbf{x}_t \in X_{test}, \quad \tilde{h}(\mathbf{x}_t) = \tilde{\boldsymbol{\beta}}^T \phi(\mathbf{x}_t)$

---

to Definition 3.1, that is, $s_c(\mathbf{u_i}) := \lambda_i (\boldsymbol{\beta}_c^\top \mathbf{u}_i)^2$ where $(\lambda_i, \mathbf{u}_i)$ is the $i$-th pair of eigenvalue and eigenvector. We then select the top-$K$ most robust features for each class $c \in \mathcal{Y}$ based on the robustness score denoted by $\tilde{\mathbf{U}}_c := \{\mathbf{u}_{\sigma(i)} \,|\, s_c(\mathbf{u}_{\sigma(i)}) \geq s_c(\mathbf{u}_{\sigma(j)}), \forall i, j \in [1, \dots, K]\}$. The global robust features for the model $\tilde{\mathbf{U}}$ is obtained as a union of the sets of classwise robust features $\tilde{\mathbf{U}}_c$. Finally, the prediction on a test data $\mathbf{x}_t$ is subsequently obtained as

$$\tilde{h}(\mathbf{x}_t) = \tilde{\boldsymbol{\beta}}^\top \phi(\mathbf{x}_t), \quad \tilde{\boldsymbol{\beta}} := \tilde{\mathbf{U}} \tilde{\mathbf{U}}^\top \boldsymbol{\beta}, \quad \tilde{\mathbf{U}} := \bigcup_{c \in \mathcal{Y}} \tilde{\mathbf{U}}_c,$$

where $\bigcup$ denotes union of sets. Therefore, the new prediction is based on the updated parameters $\tilde{\boldsymbol{\beta}}$ instead of the original $\boldsymbol{\beta}$. It is not difficult to see that this corresponds to applying the original parameters $\boldsymbol{\beta}$ on the robustly useful features, i.e., $\tilde{h}(\mathbf{x}_t) = \tilde{\boldsymbol{\beta}}^\top \phi(\mathbf{x}_t) = \boldsymbol{\beta}^\top \tilde{\mathbf{U}} \tilde{\mathbf{U}}^\top \phi(\mathbf{x}_t) = \boldsymbol{\beta}^\top \tilde{\phi}(\mathbf{x}_t)$ where $\tilde{\phi}(\mathbf{x}_t) := \tilde{\mathbf{U}} \tilde{\mathbf{U}}^\top \phi(\mathbf{x}_t)$. Figure 1 and Algorithm 1 summarize the proposed test-time defense.

### 3.3 Robustness vs information of features

We show that the robust features are also the informative features by defining a notion of informative features, inspired by usefulness property in Ilyas et al. (2019).

**Definition 3.5** (Informative features). Given a distribution $\mathcal{D}$ on $\mathcal{X} \times \mathbb{R}^C$ and a trained model $h(\mathbf{x}) = \boldsymbol{\beta}^\top \phi(\mathbf{x})$, we define the information in a feature $f$ with respect to $c$-th class component of $y \in \mathbb{R}^C$ as $\rho_{\mathcal{D}, \boldsymbol{\beta}, c}(f) = \mathbb{E}_{(\mathbf{x}, y) \sim \mathcal{D}} [y_c \boldsymbol{\beta}_c^\top f(\mathbf{x})]$, where $c \in \{1, \dots, C\}$ and $\boldsymbol{\beta}_c$ is $c$-th column of $\boldsymbol{\beta}$.

While the informative feature is similar to the $\rho-$useful feature defined in Ilyas et al. (2019), it is important to note that we define it class-specific for the penultimate feature $f$. Additionally, our definition also includes useful, non-robust features defined in Ilyas et al. (2019).

**Corollary 3.6.** *Let $(\lambda_i, \mathbf{u}_i)_{i=1,2,\dots}$ denote the eigenpairs of $\Sigma = \mathbb{E}_{\mathbf{x}} [\phi(\mathbf{x}) \phi(\mathbf{x})^\top]$. For any feature $f \in \mathcal{F}$ of the $f = \tilde{\mathbf{U}} \tilde{\mathbf{U}}^\top \phi$, where $\tilde{\mathbf{U}} = [\mathbf{u}_1 \mathbf{u}_2 \dots \mathbf{u}_K]$ is a matrix of any $K$ orthonormal eigenvectors of $\Sigma$, then information in feature $f$ with respect to $c$-th component is given by*

$$\rho_{\mathcal{D}, \boldsymbol{\beta}, c}(f) = \sum_{i=1}^K \lambda_i (\boldsymbol{\beta}_c^\top \mathbf{u}_i)^2 = \sum_{i=1}^K s_c(\mathbf{u}_i).$$

Hence, the set of features selected in Algorithm 1 by sorting the robustness score $s_c(\mathbf{u}_i)$ also correspond to the eigenvectors with the most information. However, note that to maximize $\rho_{\mathcal{D}, \boldsymbol{\beta}, c}(f)$, the full eigenspace has to be chosen, that is $K = p$. We also provide visualizations of the defined features in B.13 of Appendix.

# 4 Experimental Results

We present the following experimental analysis of RFI in this section: (1) evaluation of RFI against adaptive attacks resulting in consistent improvement in the robust performance in Section 4.1; (2) transfer attack evaluation of RFI showing the strength of RFI as well as establishing that RFI does not result in gradient obfuscation in Section 4.2; (3) in Section 4.3 we adapt static RFI to a dynamic adaptive test-time defense and show that static RFI is better than dynamic RFI. Consequently, we compare static RFI to other test-time defenses in Section 4.4 showing RFI outperforms other dynamic test-time defenses; (4) we discuss the abalations on RFI in Section 4.5. Additionally, we present the performance of RFI on calibrated models using temperature scaling (Guo et al., 2017) in the appendix since it has been shown to improve robustness (Qin et al., 2021; Grabinski et al., 2022; Stutz et al., 2020; Tao et al., 2023).

**Datasets & Resources.** We evaluate RFI on CIFAR-10, CIFAR-100 (Krizhevsky et al., 2009), Robust CIFAR-10 (Ilyas et al., 2019), tiny ImageNet (Le & Yang, 2015) and ImageNet (Russakovsky et al., 2015) datasets. We use Pytorch Paszke et al. (2019) for all our experiments & a single Nvidia DGX A100 to run all of our experiments. We also open-source our implementation at `https://github.com/Anurag14/RFI`.

**Adversarial Attacks.** We evaluate RFI on different white and black-box adversarial attacks namely, *Projected Gradient Descent (PGD)* Madry et al. (2018), a white-box attack that perturbs the input within a small $\ell_p$ radius $\epsilon$, so that it maximizes the loss of a model. We perform both $\ell_\infty$ and $\ell_2$ PGD attack with standard perturbation $\epsilon$, attack step size and iteration for each dataset. *AutoAttack* (Croce et al., 2020), a suite of white-box and black-box attacks including Auto PGD-Cross Entropy (APGD-CE), Auto PGD-Difference Logit Ratios (APGD-DLR) (Croce & Hein, 2020b), Fast Adaptive Boundary Attack (FAB) (Croce & Hein, 2020a), and Square Attacks (Andriushchenko et al., 2020). *APGD-CE and APGD-DLR* are parameter-free white-box attacks that are extensions of PGD attack with no step size parameter and stronger than PGD. *FAB* is a white-box attack that minimizes the norm of the adversarial perturbation. *Square Attack* is an efficient black-box attack that is score based and uses random search without gradient approximations. While *adaptive attacks* generate the adversarial images using the target model, *transfer attacks* generate adversarial images using a surrogate model and attack the target model.

**Benchmarking on SoTA defenses.** We evaluate RFI on various architectures trained differently: *ResNet-18* and *ResNet-50* with standard training and the popular adversarial training methods such as *PGD* Madry et al. (2018), *Interpolated Adversarial Training (IAT)* Lamb et al. (2019), *Carlini-Wagner (C&W)* loss Carlini & Wagner (2017) and *TRADES* (Zhang et al., 2019). We select different *state-of-the-art adversarially trained models* from RobustBench upon which at the test time we integrate RFI (Carmon et al., 2019; Engstrom et al., 2019; Rice et al., 2020; Wang et al., 2023; Pang et al., 2022). These methods either use additional data (Carmon et al., 2019; Wang et al., 2023), informed adversarial prior (Engstrom et al., 2019) or early stopping (Rice et al., 2020) to improve the robustness of models. We detail each training method in appendix.

**Evaluation measures.** We measure the performance of models with and without RFI by the accuracy of predictions to both clean/original samples and adversarial samples averaged over 5 runs. We use 'Clean' to denote accuracy of models to original samples. Note that there are no standard deviation in our evaluation of models from RobustBench as we are directly loading the models without training, hence no stochasticity. Details of the model evaluation are in Appendix B.2. We further remark that our defense strategy does not circumvent gradient based attacks due to gradient masking (Athalye et al., 2018b) since we simply project the last layer feature in its covariance eigenspace, hence the network remains differentiable with active gradients.

**Comparison to adaptive test-time defenses.** We compare RFI with two adaptive test-time defenses: *SODEF* (Kang et al., 2021) and *Anti-adv* (Alfarra et al., 2022). The choice of SODEF and Anti-adv is due to their relatively faster inference costs 2× and 8×, respectively, and are representative of model adaptation and input modification strategies for adaptive test-time defenses, respectively.

## 4.1 RFI improves adversarial robustness consistently

We evaluate RFI for adaptive attacks by generating adversarial samples to specifically target our defense (Tramer et al., 2020). We obtain clean and robust accuracy for standard and adversarially trained models before and after integrating RFI and setting $K$ to the number of classes. The results for different

Table 1: **Adaptive attack performance of RFI.** We consider $\ell_\infty$ and $\ell_2$ PGD attack on CIFAR-10 with Resnet-18 and $\ell_\infty$ attack with step size $\epsilon/4$ and 40 iterations. $\ell_2$ attack with size $\epsilon/5$ and 100 iterations. RFI improves the performance on an average by **2%**.

| Training | Clean | | | $\ell_\infty(\epsilon = \frac{8}{255})$ | | | $\ell_2(\epsilon = 0.5)$ | | |
|---|---|---|---|---|---|---|---|---|---|
| | Method | +RFI | % Gain | Method | +RFI | % Gain | Method | +RFI | % Gain |
| Standard | $95.28_{\pm 0.04}$ | $88.53_{\pm 0.04}$ | **-6.75** | $1.02_{\pm 0.12}$ | $4.35_{\pm 0.08}$ | **+3.33** | $0.39_{\pm 0.00}$ | $9.73_{\pm 0.10}$ | **+9.34** |
| Robust CIFAR-10 | $78.69_{\pm 0.01}$ | $\mathbf{78.75}_{\pm 0.02}$ | **+0.06** | $1.30_{\pm 0.09}$ | $7.01_{\pm 0.10}$ | **+5.71** | $9.63_{\pm 0.15}$ | $\mathbf{11.00}_{\pm 0.14}$ | **+1.37** |
| PGD | $\mathbf{83.53}_{\pm 0.01}$ | $83.22_{\pm 0.02}$ | **-0.31** | $42.20_{\pm 0.00}$ | $43.29_{\pm 0.00}$ | **+1.09** | $54.61_{\pm 0.00}$ | $\mathbf{55.03}_{\pm 0.00}$ | **+0.42** |
| IAT | $\mathbf{91.86}_{\pm 0.01}$ | $91.26_{\pm 0.00}$ | **-0.60** | $44.76_{\pm 0.03}$ | $46.95_{\pm 0.00}$ | **+2.19** | $62.53_{\pm 0.01}$ | $\mathbf{64.31}_{\pm 0.01}$ | **+1.78** |
| C&W | $\mathbf{85.16}_{\pm 0.12}$ | $84.91_{\pm 0.16}$ | **-0.25** | $40.12_{\pm 0.16}$ | $42.33_{\pm 0.32}$ | **+2.21** | $55.18_{\pm 0.28}$ | $\mathbf{56.68}_{\pm 0.30}$ | **+1.50** |
| TRADES | $\mathbf{81.22}_{\pm 0.21}$ | $80.68_{\pm 0.38}$ | **-0.54** | $51.93_{\pm 0.25}$ | $53.50_{\pm 0.27}$ | **+1.57** | $59.87_{\pm 0.36}$ | $\mathbf{61.27}_{\pm 0.44}$ | **+1.40** |

training procedures on CIFAR-10 with ResNet-18 are presented in Table 1 (CIFAR-100 with ResNet-18 and tiny ImageNet with ResNet-50 in Tables 11 and 12, respectively, in Appendix). Robust CIFAR-10 denotes standard training using Robust CIFAR-10 dataset. We observe that *our method consistently improves the robust performance of adversarially trained models, on an average by* 2%. There is a minor drop in the clean performance as we choose only a subset of the informative features that are also robust ($K \neq p$), hence a loss in information to achieve the best possible clean performance as derived in Corollary 3.6. Nevertheless, the gain in robust performance is with *almost no computational overhead*. The seemingly small improvement in performance is mainly due to the fact that we are adapting the trained model without any further learning, as well as the adaptive attacks on RFI results in a stronger adversary than the base model as we discuss in transfer attack evaluation subsequently. Additional experiments showing the effectiveness of RFI on Expectation Over Transformation attack (Athalye et al., 2018b) is presented in Table 9 of Appendix. Furthermore, RFI improves the robustness of calibrated models by $4\% - 8\%$ as shown in Tables 10, 11 and 12 of Appendix.

### 4.2 Transfer Attack Evaluation: RFI is stronger than base model

Many defences show remarkable robustness to adaptive attacks by obfuscating gradients, thereby circumventing gradient-based attacks and offering a false sense of security (Athalye et al., 2018a; Huang et al., 2021). Therefore to validate the true effectiveness of a defense, evaluating transfer attack is crucial. Hence, we expand our evaluation from Table 1 to transfer attacks, where we assess the performance with and without RFI against adversarial samples generated from the base and base model+RFI. The results in Table 2 shows that *RFI is more robust to attacks from base model whereas the base model loses considerable robustness when attacked with the adversary from RFI demonstrating that RFI is a stronger adversary than the base model*. It is interesting to note that the robustness of C&W trained model is completely lost when tested against adversarial examples from C&W+RFI model. This clearly establishes that the *RFI is not resulting in gradient obfuscation as C&W is not a gradient based attack*. Contrastingly, TRADES results in a more robust model that withstands attack from TRADES+RFI. While the performance of TRADES is almost the same for adversarial attacks generated from TRADES and TRADES+RFI, RFI results in more robust models in both cases. We present the results on calibrated models in Tables 15 and 16 in Appendix.

Table 2: **Transfer attack on ResNet-18 for CIFAR-10.** Setting same as Table 1. RFI results in much stronger adversary than the base method.

| Adversarial Examples are generated from **base model** | | | | |
|---|---|---|---|---|
| Training | $\ell_\infty(\epsilon = \frac{8}{255})$ | | $\ell_2(\epsilon = 0.5)$ | |
| | Method | +RFI | Method | +RFI |
| Standard | 1.02 | **10.36** | 0.39 | **12.09** |
| Robust CIFAR-10 | 1.30 | **15.41** | 9.63 | **17.38** |
| PGD | 42.20 | **46.02** | 54.61 | **58.81** |
| IAT | 44.76 | **49.06** | 62.53 | **66.67** |
| C&W | 40.01 | **45.48** | 55.02 | **58.95** |
| TRADES | 51.98 | **54.33** | 60.03 | **65.23** |

| Adversarial Examples are generated from **base model+RFI** | | | | |
|---|---|---|---|---|
| Training | $\ell_\infty(\epsilon = \frac{8}{255})$ | | $\ell_2(\epsilon = 0.5)$ | |
| | Method | +RFI | Method | +RFI |
| Standard | 0.00 | **4.35** | 0.01 | **9.39** |
| Robust CIFAR-10 | 0.03 | **7.01** | 1.05 | **11.00** |
| PGD | 34.80 | **43.29** | 49.90 | **55.03** |
| IAT | 35.78 | **46.95** | 55.42 | **64.31** |
| C&W | 3.92 | **42.56** | 13.50 | **56.79** |
| TRADES | 51.70 | **53.45** | 58.09 | **61.39** |

### 4.3 Static RFI is better than Dynamic/Adaptive RFI

The principle of RFI can be effectively used to adapt the model at test-time to every input by computing the transformation matrix $\tilde{U}$ using the robust feature score $s(u)$ of eigenvectors of *test set* feature covariance $\Sigma_{test}$. The results for this adaptive strategy is in Table 3 evaluated for the robust training settings of Table 1. We consider transfer attack using the base model for fair comparison and observe that *static RFI is better than dynamic RFI*. This reinforces the theoretical result that the eigendirections of the training set feature covariance determines the most robust features (Corollary 3.4). Moreover, adaptive attacks in dynamic RFI needs further information on when to adapt since the model should be static until the attacker creates an adversarial sample, and the adaptive transformation using $\tilde{U}$ should be done only in the case of defender. Details of the challenges in deploying dynamic RFI when the use case is unknown, and the results for adaptive attacks are in Table 17 in Appendix B.10.

Table 3: **Comparison of static and dynamic/adaptive RFI.** Setting same as Table 1. Adversarial examples are generated from the base model for fair comparison.

| Training | Clean | | $\ell_\infty(\epsilon = \frac{8}{255})$ | | $\ell_2(\epsilon = 0.5)$ | |
|---|---|---|---|---|---|---|
| | Static | Dynamic | Static | Dynamic | Static | Dynamic |
| PGD | 83.22 | 82.86 | 46.02 | **46.83** | 58.81 | **59.23** |
| IAT | 91.26 | **91.35** | **49.06** | 48.53 | **66.67** | 66.28 |
| C&W | 84.97 | 83.01 | **45.48** | 43.98 | **58.95** | 57.82 |
| TRADES | 80.76 | 78.98 | **54.33** | 53.58 | **65.23** | 65.00 |

### 4.4 Static RFI outperforms adaptive test-time defenses

As a result of the static vs dynamic RFI evaluation in Section 4.3, we compare the effectiveness of static RFI on both white-box and black-box attacks with other adaptive test-time defenses such as SODEF and Anti-adv. In Table 4, we present the result for APGD-CE, APGD-DLR, FAB, Square and AutoAttack for different SOTA methods. We observe that *adding static RFI improves the performance across all the methods*. Importantly, RFI despite being non-adaptive improves the robust performance (at least marginally) for all the SOTA methods, even though each one of them achieves robustness by incorporating very different strategies like additional data, early stopping or informed prior. While this shows the strength and effectiveness of RFI, it also raises a fundamental question of whether it is necessary to adapt the model to individual test samples in order to improve the robustness in the adaptive test-time defense strategy. Evaluation of other SOTA models for CIFAR-10, CIFAR-100 and ImageNet showing similar observations are in Table 13 in Appendix.

Table 4: **Robust performance evaluation of RFI on state-of-the-art methods.** We evaluate APGD-CE, APGD-DLR, FAB, Square and AutoAttack under $\ell_\infty(\epsilon = 8/255)$ on CIFAR-10 and CIFAR-100. RFI consistently **improves** the performance of the base model, whereas Anti-adv and SODEF results in slight **decrease** in the performance to AutoAttack. The inference time of RFI is $1\times$, whereas Anti-adv (Alfarra et al., 2022) and SODEF (Kang et al., 2021) are $8\times$ and $2\times$, respectively. Additional results are in Table 13 in Appendix. There is no standard deviation as the trained models are from RobustBench.

| | Base Method | Defense | Clean | APGD-CE | APGD-DLR | FAB | Square | AutoAttack |
|---|---|---|---|---|---|---|---|---|
| CIFAR-10 | Carmon et al. (2019) WideResNet-28-10 | None | **89.69** | 61.82 | 60.85 | 60.18 | 66.51 | 59.53 |
| | | Anti-adv | **89.69** | 61.81 | 60.89 | 60.11 | 66.58 | **58.70** |
| | | SODEF | 89.68 | 60.20 | 60.72 | 58.04 | 65.28 | **57.23** |
| | | RFI ($K = 10$) | 89.60 | 62.38 | 61.58 | 60.21 | 66.59 | 60.72 |
| | | RFI (opt. $K = 20$) | 89.60 | **62.45** | **61.60** | **60.38** | **66.90** | **61.02** |
| CIFAR-100 | Pang et al. (2022) WideResNet-28-10 | None | **63.66** | 35.29 | 31.71 | 31.32 | 35.70 | 31.08 |
| | | Anti-adv | 63.41 | 32.50 | 30.32 | 31.30 | 35.76 | **30.10** |
| | | SODEF | 63.08 | 30.96 | 29.54 | 31.44 | 32.27 | **30.56** |
| | | RFI ($K = 100$) | 63.01 | 36.03 | **31.95** | 31.88 | 35.79 | 31.29 |
| | | RFI (opt. $K = 115$) | 63.10 | **36.07** | **31.95** | **31.96** | **35.88** | **31.91** |

We compare the algorithmic time complexity for different test-time defenses in Tables 4 and 13 and provide the average time to infer a single sample on a Nvidia DGX-A100 in the Table 5. Note that the average time closely follows the time complexity. RFI does not add additional computation overhead. However, Anti-adv and SODEF lead to 8× and 2× computation compared to the base model.

Table 5: **Time comparison** for RFI, Anti-adv, SODEF. RFI: 1×, Anti-adv: 8×, SODEF: 2×.

| Model | Time Comparison in (ms) | | | |
|---|---|---|---|---|
| | Base | RFI | Anti-adv | SODEF |
| PreActResnet-18 | 0.2760 | **0.2777** | 1.5127 | 0.5133 |
| ResNet-50 | 0.3692 | **0.3703** | 2.7684 | 0.6877 |
| WideResnet-28-10 | 0.3780 | **0.3763** | 2.9735 | 0.7338 |
| WideResnet-34-10 | 0.8619 | **0.8654** | 6.8380 | 1.6599 |

### 4.5  Abalation Studies

### 4.5.1  Effect of adversary strength

We study the effect of adversary strength on our method, RFI, by taking the adversarially trained ResNet-18 on CIFAR-10 using PGD ($\epsilon = 8/255$ for $\ell_\infty$ and $\epsilon = 0.5$ for $\ell_2$) as the base model. Table 6 shows the evaluation of RFI with $\ell_\infty$ PGD attack for $\epsilon = \{2/255, 4/255, 12/255, 16/255\}$ and 40 iterations, and $\ell_2$ attack for $\epsilon = \{0.25, 0.75, 1\}$ and 100 iterations. The results show that *the underlying model augmented with RFI consistently improves over baseline across the perturbations of various strengths*, especially by over 1% for adversary that is stronger than the base model ($\epsilon = \{12/255, 16/255\}$ for $\ell_\infty$ and $\epsilon = \{0.75, 1.00\}$ for $\ell_2$.

Table 6: **RFI consistently improves over baseline across the perturbations of various strengths.** Evaluation of RFI for $\ell_\infty$ and $\ell_2$ on ResNet-18 adversarially trained with CIFAR-10 and PGD.

| Method | $\ell_\infty$ attack | | | | $\ell_2$ attack | | |
|---|---|---|---|---|---|---|---|
| | $\epsilon = \frac{2}{255}$ | $\epsilon = \frac{4}{255}$ | $\epsilon = \frac{12}{255}$ | $\epsilon = \frac{16}{255}$ | $\epsilon = 0.25$ | $\epsilon = 0.75$ | $\epsilon = 1.00$ |
| PGD | 74.60 | 64.02 | 23.34 | 11.66 | 71.34 | 40.91 | 28.25 |
| PGD+RFI | **74.99** | **64.91** | **24.32** | **12.55** | **71.48** | **41.95** | **29.24** |

Further empirical analysis of the effect of step size in PGD attack is provided in Table 19 in Appendix.

### 4.5.2  Choice of $K$

To study the effect of parameter $K$ in detail, we vary $K$ for the adversarial training methods on CIFAR-10 with ResNet-18 under $\ell_\infty(\epsilon = 8/255)$ threat model, same setting as in Table 1 setting. Figure 2 (left plot) shows that the adversarial training methods (PGD, IAT, C&W and TRADES) behave similarly in their accuracy profile as compared to standard training even on Robust CIFAR-10 dataset. Moreover, the best performance is for $K = 10$ for all the robust training methods. The corresponding eigenvalue spectrum exhibits a knee drop after top-10 eigenvalues (right plot), which motivates *our choice of $K$ as top-10 features for each class, equivalent to the number of classes.* As a complementary explanation for our choice of $K$, neural collapse phenomenon observes that the penultimate feature of each class collapses to its mean after the training error is almost zero (Papyan et al., 2020). This implies that there is principally only $C$ number of feature vectors, one for each class, justifying our choice. Further ablation studies on the effect of parameter $K$ for SoTA models are in Figures 4 and 5 in Appendix. While we set $K$ to be the number of classes, we also report the best performance of RFI by finding the optimal $K$ using grid search for SoTA models in Table 13 in Appendix. Although $K =$ number of classes is not the optimum for the SoTA models, it is still better than SODEF and Anti-adv.

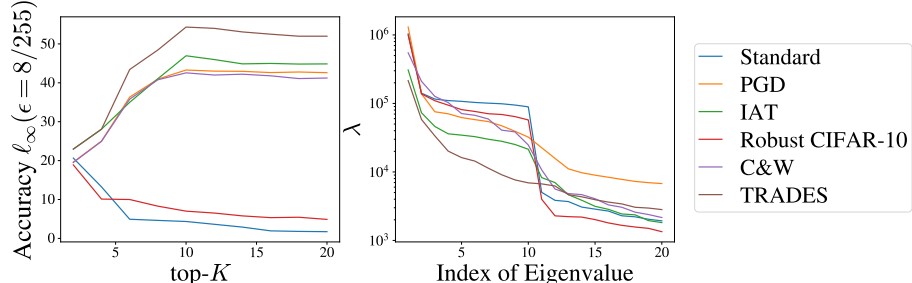

Figure 2: **Effect of $K$ in RFI.** Robust accuracy and eigenvalue profile in ascending order of all the methods in Table 10.

### 4.5.3 Comparison of RFI to similar conceptual methods

The conceptual counterparts to RFI include performing the projection of intermediary layers to a low dimensional space instead of the last layer, or enforcing low dimensional last layer directly. In Table 7, we evaluate effectiveness of performing RFI on intermediate layers by truncating the last but one hidden layer of ResNet-18 and evaluate the PGD trained model considered in Table 1. This hidden layer has $512 \times 4 \times 4$ convolution which we project to $10 \times 4 \times 4$ using RFI procedure. *While it is clear that performing RFI on the last layer as derived theoretically improves the robust performance, RFI on intermediary layers harm the robustness.* We provide the result for enforcing low dimension last layer in appendix (Table 20) which also demonstrates the superiority of RFI.

Table 7: **RFI on last layer outperforms intermediate layer.** Evaluation of PGD trained ResNet-18 on CIFAR-10.

| None | RFI on last layer | RFI on last but one layer |
|---|---|---|
| 42.20 | **43.29** | 36.06 |

## 5   Discussion

The simplicity and effectiveness of RFI at test-time is impressive as the robustness gain is achieved with zero additional computation overhead for inference. While RFI on smaller models like ResNet demonstrate more improvement in robustness than larger SoTA models like WideResNet, it is important to note that these SoTA models are already optimized to their full potential, hence even a small improvement is significant in these cases. Furthermore, RFI is well-founded theoretically. Consequently, the idea of RFI can also be used to develop a robust training procedure by incorporating the projection onto the robust feature space during training. We leave the experimental analysis for future study as the current work focuses on test-time defenses. However, it is intriguing to theoretically analyze the robustness of features during training to understand the RFI's potential as an idea and the cause of vulnerability to adversarial examples. Therefore, we derive the learning dynamics of full batch gradient descent on population squared error loss of GAM (stated informally in Proposition 5.1 and proved in Appendix A.5).

**Proposition 5.1** (Learning dynamics of GAM). *Given $h(\boldsymbol{x}) = \boldsymbol{\beta}^\top \phi(\boldsymbol{x})$ and $\Sigma = \mathbb{E}_{\mathbf{x}} \left[ \phi(\mathbf{x}) \phi(\mathbf{x})^\top \right]$. Let $(\lambda_i, \boldsymbol{u}_i)$ be the eigenpair of $\Sigma$. Then full batch gradient descent learns features in the direction of $\boldsymbol{u}_i$ with large eigenvalues $\lambda_i$ first during the training and those directions are robust only if they align with the original signal direction $\boldsymbol{\beta}$.*

This result further strengthens the idea and suggests that truncating the non-robust directions during training, which is one of the plausible causes for the existence of adversaries, could improve the robustness of the model.

**Connection to Neural Tangent Kernel (NTK) features.** One of the related results to Proposition 5.1 is using the NTK features. Tsilivis & Kempe (2022) defined features using NTK gram matrix and empirically

observed that the features corresponding to the top spectrum of NTK are more robust and learned first during training. Our theoretical framework enables us to establish the equivalence of NTK features to the robust feature definition and more importantly prove that the robust NTK features indeed correspond to the top of the spectrum. The NTK gram matrix $\boldsymbol{\Theta} \in \mathbb{R}^{n \times n}$ is between all pairs of datapoints. NTK features of input $\boldsymbol{x}$ is defined using the eigendecomposition of $\boldsymbol{\Theta} = \sum_{i=1}^{n} \lambda_i \mathbf{v_i} \mathbf{v_i}^T$ as $f_i^{ker}(\boldsymbol{x}) := g(\lambda_i, \boldsymbol{v}_i, \boldsymbol{x})$ for a specific function $g$. We state the result in the following proposition and prove along with empirical verification in Appendix A.6 (Figure 3).

**Proposition 5.2** (NTK feature robustness lies at the top). *Let feature $f_i^{ker}$ be Lipschitz continuous in gradient of NTK with respect to $\mathbf{x}$ and an adversarial perturbation $\boldsymbol{\delta}$ such that $||\boldsymbol{\delta}||_p \leq \Delta$. Then, $||f_i^{ker}(\mathbf{x} + \boldsymbol{\delta}) - f_i^{ker}(\mathbf{x})||_2 \leq \Theta(\frac{1}{\lambda_i})$.*

Although we prove that the robust NTK features correspond to the top of the spectrum, we leave the challenge to establish its connection to the DNN for future analysis. Overall, our work develops a guaranteed algorithm to improve adversarial robustness at test-time along with possibilities to improve the robust training procedures. Additionally, we note that while the current work focused on evaluating SOTA convolution-based models, the effectiveness of RFI on transformer-based models is an interesting future direction.

## 6 Conclusion

In this paper, we present a novel test-time defense that can be seamlessly integrated with any method at the time of deployment to improve the robustness of the underlying model. While the adaptive test-time defense as an approach offers promise to improve the robustness of models at the deployment stage, the general criticism of available methods is that they significantly increase the inference time of the underlying model. Our method, Robust Feature Inference (RFI), has no effect on the inference time of the underlying model which makes it a practical alternative for adaptive test-time defense. We also present a comprehensive theoretical justification for our approach describing the motivation behind retaining features in the top eigenspectrum of the feature covariance. In addition, we show that these top features are more robust and informative, and validate our algorithm through extensive experiments. In conclusion, we propose the first theoretically guided adaptive test-time defense algorithm that has the same inference time as the base model with significant experimental results. Our findings contribute to the ongoing efforts to develop robust models that can resist adversarial examples and improve the security and reliability of DNNs.

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

# A Proofs of the Main Results

In this section, we prove Theorem 3.2, and related results, Corollaries 3.4–3.6 and Remark 3.3.

## A.1 Proof of Theorem 3.2

*Proof.* Recall that we assume $y = h(\mathbf{x}) + \boldsymbol{\epsilon} = \boldsymbol{\beta}^\top \phi(\mathbf{x}) + \boldsymbol{\epsilon}$, where $\boldsymbol{\epsilon} \in \mathbb{R}^C$ has independent coordinates, each satisfying $\mathbb{E}[\epsilon_c] = 0$, $\mathbb{E}[\epsilon_c^2] \leq \sigma^2$ for all $c \in \{1, \dots, C\}$. The features for which we wish to compute robustness are of the form $f = \boldsymbol{M}\phi$ where $\boldsymbol{M}$ is a linear map.

We are interested in robustness with respect to the $c$-th component, which is computed as

$$
\begin{aligned}
s_{\mathcal{D},\boldsymbol{\beta},c}(f) &= \mathbb{E}_{(\mathbf{x},y)\sim\mathcal{D}}\left[\inf_{||\tilde{\mathbf{x}}-\mathbf{x}||_2 \leq \Delta} y_c \boldsymbol{\beta}_c^\top f(\tilde{\mathbf{x}})\right] \\
&= \mathbb{E}_{(\mathbf{x},y)\sim\mathcal{D}}\left[y_c \boldsymbol{\beta}_c^\top f(\mathbf{x})\right] + \mathbb{E}_{(\mathbf{x},y)\sim\mathcal{D}}\left[\inf_{||\tilde{\mathbf{x}}-\mathbf{x}||_2 \leq \Delta} y_c \boldsymbol{\beta}_c^\top \left(f(\tilde{\mathbf{x}}) - f(\mathbf{x})\right)\right]
\end{aligned}
\tag{1}
$$

We compute the first term exactly as

$$
\begin{aligned}
\mathbb{E}_{(\mathbf{x},y)\sim\mathcal{D}}\left[y_c \boldsymbol{\beta}_c^\top f(\mathbf{x})\right] &= \mathbb{E}_{\mathbf{x},\epsilon_c}\left[(\boldsymbol{\beta}_c^\top \phi(\mathbf{x}) + \epsilon_c)\boldsymbol{\beta}_c^\top f(\mathbf{x})\right] && \text{(since } \mathbb{E}[\epsilon_c] = 0\text{),} \\
&= \mathbb{E}_{\mathbf{x}}\left[\boldsymbol{\beta}_c^\top \phi(\mathbf{x})\phi(\mathbf{x})^\top \boldsymbol{M}\boldsymbol{\beta}_c\right] \\
&= \boldsymbol{\beta}_c^\top \Sigma \boldsymbol{M}\boldsymbol{\beta}_c, && (\Sigma = \mathbb{E}_{\mathbf{x}}\left[\phi(\mathbf{x})\phi(\mathbf{x})^\top\right]).
\end{aligned}
$$

For the second term in (1), we aim to derive a lower bound. Observe that

$$
\begin{aligned}
y_c \boldsymbol{\beta}_c^\top \left(f(\tilde{\mathbf{x}}) - f(\mathbf{x})\right) &= y_c \boldsymbol{\beta}_c^\top \boldsymbol{M}\left(\phi(\tilde{\mathbf{x}}) - \phi(\mathbf{x})\right) \\
&\geq -|y_c| \cdot \|\boldsymbol{\beta}_c\|_{\mathcal{H}} \cdot \|\boldsymbol{M}\left(\phi(\tilde{\mathbf{x}}) - \phi(\mathbf{x})\right)\|_{\mathcal{H}} \\
&\geq -|y_c| \cdot \|\boldsymbol{\beta}_c\|_{\mathcal{H}} \cdot \|\boldsymbol{M}\|_{op} \cdot \|\phi(\tilde{\mathbf{x}}) - \phi(\mathbf{x})\|_{\mathcal{H}}.
\end{aligned}
$$

Using $L$-Lipschitzness of $\phi$, we have $\|\phi(\tilde{\mathbf{x}}) - \phi(\mathbf{x})\|_{\mathcal{H}} \leq L\|\tilde{\mathbf{x}} - \mathbf{x}\|_2 \leq L\Delta$. Hence, the second term in (1) can be bounded from below as

$$
\begin{aligned}
\mathbb{E}_{(\mathbf{x},y)\sim\mathcal{D}}\left[\inf_{||\tilde{\mathbf{x}}-\mathbf{x}||_2 \leq \Delta} y_c \boldsymbol{\beta}_c^\top \left(f(\tilde{\mathbf{x}}) - f(\mathbf{x})\right)\right] &= -\|\boldsymbol{M}\|_{op} \cdot \|\boldsymbol{\beta}_c\|_{\mathcal{H}} \cdot L\Delta \cdot \mathbb{E}_{\mathbf{x},\epsilon_c}\left[|y_c|\right] \\
&\geq -\|\boldsymbol{M}\|_{op} \cdot \|\boldsymbol{\beta}_c\|_{\mathcal{H}} \cdot L\Delta \cdot \mathbb{E}_{\mathbf{x},\epsilon_c}\left[|\boldsymbol{\beta}_c^\top \phi(\mathbf{x}) + \epsilon_c|\right]
\end{aligned}
$$

Finally, using Jensen's inequality, we can write

$$
\mathbb{E}_{\mathbf{x},\epsilon_c}\left[|\boldsymbol{\beta}_c^\top \phi(\mathbf{x}) + \epsilon_c|\right] \leq \sqrt{\mathbb{E}_{\mathbf{x},\epsilon_c}\left[(\boldsymbol{\beta}_c^\top \phi(\mathbf{x}) + \epsilon_c)^2\right]} \leq \sqrt{\sigma^2 + \boldsymbol{\beta}_c^\top \Sigma \boldsymbol{\beta}_c}.
$$

Combining the above computation leads to

$$
s_{\mathcal{D},\boldsymbol{\beta},c}(f) \geq \boldsymbol{\beta}_c^\top \Sigma \boldsymbol{M}\boldsymbol{\beta}_c - L\Delta\|\boldsymbol{M}\|_{op}\|\boldsymbol{\beta}_c\|_{\mathcal{H}}\sqrt{\sigma^2 + \boldsymbol{\beta}_c^\top \Sigma \boldsymbol{\beta}_c},
$$

which proves Theorem 3.2. $\qquad\square$

## A.2 Proof of Remark 3.3

*Proof.* The proof requires assumption of a Gaussian model, i.e., $\mathbf{x} \sim \mathcal{N}(0, \Sigma)$ and $\epsilon_c \sim \mathcal{N}(0, \sigma^2)$. Since the feature map is assumed to be linear, $\phi(\mathbf{x}) = \mathbf{x}$, it follows that $y_c = \boldsymbol{\beta}_c^\top \mathbf{x} + \epsilon_c$ is also Gaussian $y_c \sim \mathcal{N}(0, \sigma^2 + \boldsymbol{\beta}_c^\top \Sigma \boldsymbol{\beta}_c)$ and hence, $|y_c|$ is half-normal distributed.

Now recall that the first term in (1) can be computed exactly as $\boldsymbol{\beta}_c^\top \Sigma \boldsymbol{M}\boldsymbol{\beta}_c$. To compute the second term in (1), note that

$$
\inf_{||\tilde{\mathbf{x}}-\mathbf{x}||_2 \leq \Delta} y_c \boldsymbol{\beta}_c^\top \left(f(\tilde{\mathbf{x}}) - f(\mathbf{x})\right) = \inf_{||\tilde{\mathbf{x}}-\mathbf{x}||_2 \leq \Delta} y_c \boldsymbol{\beta}_c^\top \boldsymbol{M}(\tilde{\mathbf{x}} - \mathbf{x})
$$

and the infimum is achieved when the difference is aligned with $\boldsymbol{M}^\top\boldsymbol{\beta}_c$, that is, $\tilde{\mathbf{x}} = \mathbf{x} \pm \Delta \frac{\boldsymbol{M}^\top\boldsymbol{\beta}_c}{\|\boldsymbol{M}^\top\boldsymbol{\beta}_c\|_{\mathcal{H}}}$. The sign depends on the sign of $y_c$, which leads to the second term in (1) compute to

$$\mathbb{E}_{(\mathbf{x},y)\sim\mathcal{D}}\left[\inf_{\|\tilde{\mathbf{x}}-\mathbf{x}\|_2\leq\Delta} y_c\boldsymbol{\beta}_c^\top\left(f(\tilde{\mathbf{x}})-f(\mathbf{x})\right)\right] = -\mathbb{E}_{\mathbf{x},\epsilon_c}\left[|y_c|\right]\cdot\Delta\cdot\|\boldsymbol{M}^\top\boldsymbol{\beta}_c\|_{\mathcal{H}}.$$

Since $|y_c|$ is half-normal, $\mathbb{E}[|y_c|] = \sqrt{2/\pi}\sqrt{\sigma^2 + \boldsymbol{\beta}_c^\top\Sigma\boldsymbol{\beta}_c}$, while $\|\boldsymbol{M}^\top\boldsymbol{\beta}_c\|_{\mathcal{H}} \leq \|\boldsymbol{M}\|_{op}\|\boldsymbol{\beta}_c\|_{\mathcal{H}}$, with the inequality being tight when $\boldsymbol{\beta}_c$ is the eigenvector of $\boldsymbol{M}$, corresponding to the largest eigenvalue. $\qquad\square$

## A.3 Proofs of Corollary 3.4 and Corollary 3.6

*Proof.* In what follows, we restrict the linear map $\boldsymbol{M}$ as $\boldsymbol{M} = \tilde{\mathbf{U}}\tilde{\mathbf{U}}^\top = \sum_{i=1}^K \mathbf{u}_i\mathbf{u}_i^\top$ where $\tilde{\mathbf{U}} = [\mathbf{u}_1,\ldots,\mathbf{u}_K]$ is an orthonormal matrix of basis for a $K$-dimensional subspace. Since the operator norm $\|\boldsymbol{M}\|_{op} = 1$ for projection matrix, the problem of finding the most robust subspace corresponds to maximising $\boldsymbol{\beta}_c^\top\Sigma\boldsymbol{M}\boldsymbol{\beta}_c = \sum_{i=1}^K \boldsymbol{\beta}_c^\top\Sigma\mathbf{u}_i\mathbf{u}_i^\top\boldsymbol{\beta}_c$.

Note that if $(\lambda,\mathbf{u})$ is an eigenpair of $\Sigma$, then $\boldsymbol{\beta}_c^\top\Sigma\mathbf{u}\mathbf{u}^\top\boldsymbol{\beta}_c = \lambda(\boldsymbol{\beta}_c^\top\mathbf{u})^2$. Hence, if we restrict the choice of $\mathbf{u}_1,\ldots,\mathbf{u}_K$ to the eigenvectors of $\Sigma$, the optimal projection is obtained by choosing the $K$ eigenvectors for which the robustness score $s_c(\mathbf{u}) = \lambda(\boldsymbol{\beta}_c^\top\mathbf{u})^2$ are largest. So the claim of Corollary 3.4 holds only if the projections are restricted to eigenspaces of $\Sigma$. The claim of Corollary 3.6 follows along the same line as the information of the feature $f = \boldsymbol{M}\phi$ can be computed as $\rho_{\mathcal{D},\boldsymbol{\beta},c}(f) = \boldsymbol{\beta}_c^\top\Sigma\boldsymbol{M}\boldsymbol{\beta}_c$. For the case of $\boldsymbol{M} = \tilde{\mathbf{U}}\tilde{\mathbf{U}}^\top$ where $\tilde{\mathbf{U}}$ is matrix of $K$ eigenvectors of $\Sigma$, we have $\rho_{\mathcal{D},\boldsymbol{\beta},c}(f) = \sum_{i=1}^K \lambda_i(\boldsymbol{\beta}_c^\top\mathbf{u}_i)^2$. Hence, if the search is restricted to eigenspaces of $\Sigma$, the most robust features also correspond to the most informative ones. $\qquad\square$

**Robust and informative features over all possible $K$-dimensional subspaces.** If we consider $\boldsymbol{M} = \tilde{\mathbf{U}}\tilde{\mathbf{U}}^\top$ for any $\tilde{\mathbf{U}} = [\mathbf{u}_1,\ldots,\mathbf{u}_K]$ with orthonormal columns, as assumed in Corollary 3.4, then

$$\boldsymbol{\beta}_c^\top\Sigma\boldsymbol{M}\boldsymbol{\beta}_c = \sum_{i=1}^K \boldsymbol{\beta}_c^\top\Sigma\mathbf{u}_i\mathbf{u}_i^\top\boldsymbol{\beta}_c = \text{Trace}\left(\tilde{\mathbf{U}}\boldsymbol{\beta}_c\boldsymbol{\beta}_c^\top\Sigma\tilde{\mathbf{U}}^\top\right) = \text{Trace}\left(\tilde{\mathbf{U}}\Sigma\boldsymbol{\beta}_c\boldsymbol{\beta}_c^\top\tilde{\mathbf{U}}^\top\right),$$

where the last equality follows from taking transpose. Hence, the resulting maximisation problem can be written as

$$\max_{\tilde{\mathbf{U}}} \boldsymbol{\beta}_c^\top\Sigma\boldsymbol{M}\boldsymbol{\beta}_c \equiv \max_{\tilde{\mathbf{U}}} \text{Trace}\left(\tilde{\mathbf{U}}\boldsymbol{\beta}_c\boldsymbol{\beta}_c^\top\Sigma\tilde{\mathbf{U}}^\top\right) \equiv \max_{\tilde{\mathbf{U}}} \text{Trace}\left(\tilde{\mathbf{U}}\mathbf{B}_c\tilde{\mathbf{U}}^\top\right), \tag{2}$$

where $\mathbf{B}_c = \frac{1}{2}(\boldsymbol{\beta}_c\boldsymbol{\beta}_c^\top\Sigma + \Sigma\boldsymbol{\beta}_c\boldsymbol{\beta}_c^\top)$. The above trace maximisation problem corresponds to finding the $K$ dominant eigenvectors of the matrix $\mathbf{B}_c$. This leads to an alternative to Algorithm 1 for finding robust projections for the test-time defense. The approach comprises of computing the dominant eigenvectors $\tilde{\mathbf{U}}_c$ of the matrix $\mathbf{B}_c$ for every class component $c \in \{1,\ldots,C\}$ and defining the robust output as $\tilde{h}(\mathbf{x}) = [\boldsymbol{\beta}_1^\top\tilde{\mathbf{U}}_1\tilde{\mathbf{U}}_1^\top\phi(\mathbf{x}),\ldots,\boldsymbol{\beta}_C^\top\tilde{\mathbf{U}}_C\tilde{\mathbf{U}}_C^\top\phi(\mathbf{x})]$. The approach would result in theoretically more robust projections, but suffers computationally since it requires $(C+1)$ eigendecompositions instead of only one eigendecomposition in Algorithm 1. Hence, it has $O(C)$ more one-time computation than Algorithm 1, but with identical inference time. The conclusion of Corollary 3.6 that the most robust features, obtained from the maximisation in (2), are also the most informative features still holds in this case.

## A.4 Dynamics of robust feature learning under GAM

In this short analysis, we argue that if the trained model is a Generalized Additive Model (GAM), $h(\mathbf{x}) = \boldsymbol{\beta}^T\phi(x)$, the test-time defense of Algorithm 1 could also be replicated through an early stopping of the training process. In other words, we argue that the components of $\boldsymbol{\beta}_c^T\phi(x)$ along the robust features—the eigen directions for which $s(\boldsymbol{u}) = \lambda(\boldsymbol{\beta}_c^\top\boldsymbol{u})^2$ are higher—are learned earlier.

For simplicity of analysis, we consider only the learning for $c$-th components, which corresponds to the following regression problem under GAM: Given training sample $\mathcal{D}_{\text{train}} := \{(\mathbf{x_i},y_i)\}_{i=1}^n \subseteq \mathcal{X}\times\mathbb{R}$, minimize the squared loss

$$\underset{\boldsymbol{b}\in\mathbb{R}^p}{\text{minimize}} \frac{1}{2n}\sum_{i=1}^n \|y_i - \boldsymbol{b}^\top\phi(\mathbf{x}_i)\|_2^2.$$

## A.5 Proof of Proposition 5.1

*Proof.* The optimal solution for $\boldsymbol{b}$ for the above problem when population squared loss is minimized is given by $\boldsymbol{\beta}_c = (\Phi\Phi^\top)^{-1}\Phi\boldsymbol{y}$, where $\Phi = [\phi(\mathbf{x}_1), \ldots, \phi(\mathbf{x}_n)]$ and $\boldsymbol{y} = [y_1 \ldots y_n]^\top$. Furthermore, if the above optimisation is solved using gradient descent with learning rate $\eta > 0$ and initialisation $\boldsymbol{b}^{(0)} = 0$, the parameters $\boldsymbol{b}^{(t)}$ are learned over the iterations as

$$\boldsymbol{b}^{(t)} = \left(I - \frac{\eta}{n}\Phi\Phi^\top\right)\boldsymbol{b}^{(t-1)} + \frac{\eta}{n}\Phi\boldsymbol{y}$$

$$= \sum_{k=0}^{t-1}\left(I - \frac{\eta}{n}\Phi\Phi^\top\right)^k \frac{\eta}{n}\Phi\boldsymbol{y}$$

$$= \sum_{k=0}^{t-1}\left(I - \frac{\eta}{n}\Phi\Phi^\top\right)^k \cdot \frac{\eta}{n}\Phi\Phi^\top\boldsymbol{\beta}_c, \tag{3}$$

with $\boldsymbol{b}^{(t)} \to \boldsymbol{\beta}_c = (\Phi\Phi^\top)^{-1}\Phi\boldsymbol{y}$ as $t \to \infty$. Suppose the eigen decomposition $\Sigma_{\text{train}} = \frac{1}{n}\Phi\Phi^\top$ is given by $\Sigma_{\text{train}} = \mathbf{U}\text{diag}(\boldsymbol{\lambda})\mathbf{U}^\top = \sum_{i=1}^{p}\lambda_i\boldsymbol{u}_i\boldsymbol{u}_i^\top$. Hence, (3) becomes

$$(3) = \sum_{k=0}^{t-1}\left(\mathbf{U}\mathbf{U}^\top - \eta\mathbf{U}\text{diag}(\boldsymbol{\lambda})\mathbf{U}^\top\right)^k \cdot \eta\mathbf{U}\text{diag}(\boldsymbol{\lambda})\mathbf{U}^\top\boldsymbol{\beta}_c$$

$$= \sum_{k=0}^{t-1}\eta\mathbf{U}(I - \eta\text{diag}(\boldsymbol{\lambda}))^k\text{diag}(\boldsymbol{\lambda})\mathbf{U}^\top\boldsymbol{\beta}_c$$

$$\boldsymbol{b}^{(t)} = \sum_{i=1}^{p}(1 - (1 - \eta\lambda_i)^t)\boldsymbol{u_i}\boldsymbol{u_i}^\top\boldsymbol{\beta}_c \tag{4}$$

From (4), it is clear that $\boldsymbol{u}_i$ directions are learnt in the order of $\lambda_i$. That is large $\lambda_i$ learned early during the training since $(1 - (1 - \eta\lambda_i)^t)$ is decreasing and at fixed $t$, the eigendirection $\boldsymbol{u}_i$ with the largest $\lambda_i$ is learned first. This proves that the direction with maximum variance is learned first. When the top eigendirection $\boldsymbol{u}_i$ aligns with the true signal $\boldsymbol{\beta}_c$, $\boldsymbol{u}_i$ will be the most robust direction as well. Hence, the top directions based on descending order of $\lambda$ is more robust if the directions align with the true underlying signal. $\square$

## A.6 Connection to Neural Tangent Kernel features

We first briefly discuss NTK and the NTK features before proving the Proposition 5.2.

**Neural Tangent Kernels (NTKs) and NTK features.** Jacot et al. (2018); Arora et al. (2019); Yang (2019) show the equivalence of training a large width neural network by gradient descent to a deterministic kernel machine called Neural Tangent Kernel. In the context of adversarial attacks and robustness, Tsilivis & Kempe (2022) propose a method to generate adversarial examples using NTK and show transferability of the attack to the finite width neural network counterpart successfully. Additionally, the authors define NTK features using the eigenspectrum of the NTK gram matrix and observe that the robust features correspond to the top of the eigenspectrum and learned first during training. In the following, we define the NTK and NTK features and show its equivalence to our robust feature definition along with the proof that the robust NTK features correspond to the top of the spectrum. The NTK gram matrix $\boldsymbol{\Theta} \in \mathbb{R}^{n\times n}$ is between all pairs of datapoints and the NTK between $\mathbf{x_i}$ and $\mathbf{x_j}$ for a network that outputs $f(\mathbf{w}, \mathbf{x})$ at data point $\mathbf{x} \in \mathbb{R}^d$ parameterized by $\mathbf{w} \in \mathbb{R}^p$ is defined by the gradient of the network with respect to $\mathbf{w}$ as

$$\boldsymbol{\Theta}(\mathbf{x_i}, \mathbf{x_j}) = \mathbb{E}_{\mathbf{w}\sim\mathcal{N}(0,\mathbf{I}_p)}[\nabla_{\mathbf{w}}f(\mathbf{w}, \mathbf{x_i})^T\nabla_{\mathbf{w}}f(\mathbf{w}, \mathbf{x_j})]. \tag{5}$$

For an extremely large width network, gradient descent optimization with least square loss is equivalent to kernel regression, the kernel being the NTK. Formally, for a data $\mathbf{x}$, the converged network output in the large width limit is $f(\mathbf{w}, \mathbf{x}) = \boldsymbol{\Theta}(\mathbf{x}, \mathbf{X})^T\boldsymbol{\Theta}(\mathbf{X}, \mathbf{X})^{-1}\mathbf{Y}$. Tsilivis & Kempe (2022) define NTK features using the eigendecomposition of $\boldsymbol{\Theta}(\mathbf{X}, \mathbf{X}) = \sum_{i=1}^{n}\lambda_i\mathbf{v_i}\mathbf{v_i}^T$ as

$$f(\mathbf{w}, \mathbf{x}) = \mathbf{\Theta}(\mathbf{x}, \mathbf{X})^T \mathbf{\Theta}(\mathbf{X}, \mathbf{X})^{-1} \mathbf{Y} = \sum_{i=1}^{n} \lambda_i^{-1} \mathbf{\Theta}(\mathbf{x}, \mathbf{X})^T \mathbf{v_i} \mathbf{v_i}^T \mathbf{Y} := \sum_{i=1}^{n} f_i^{ker}(\mathbf{x}) \tag{6}$$

where $f_i^{ker}(\mathbf{x}) \in \mathbb{R}^C$ is the $i$-th NTK feature of $\mathbf{x}$. Note that $f_i^{ker}$ is in accordance to our feature definition. We prove the empirical observation that the top spectrum-induced NTK features $f^{ker}$ are more robust in the following.

### A.7 Proof of Proposition 5.2

*Proof.* Suppose that the NTK feature $f_i^{ker}$ is $L$-Lipschitz continuous in gradient of NTK with respect to $\mathbf{x}$. Then, we have

$$\|\nabla_{\mathbf{x}} \mathbf{\Theta}(\mathbf{x} + \boldsymbol{\delta}, \mathbf{X}) - \nabla_{\mathbf{x}} \mathbf{\Theta}(\mathbf{x}, \mathbf{X})\|_2 \leq L\|\boldsymbol{\delta}\|_2. \tag{7}$$

Recall that we can write the $i$-th NTK feature as $f_i^{ker}(\mathbf{x}) := \lambda_i^{-1} \mathbf{\Theta}(\mathbf{x}, \mathbf{X})^{\top} \mathbf{v_i} \mathbf{v_i}^{\top} \mathbf{Y}$. Bounding $\|f_i^{ker}(\mathbf{x} + \boldsymbol{\delta}) - f_i^{ker}(\mathbf{x})\|_2$ by Taylor's expansion and applying (7) yield

$$
\begin{aligned}
\|f_i^{ker}(\mathbf{x} + \boldsymbol{\delta}) - f_i^{ker}(\mathbf{x})\|_2 &\overset{(a)}{=} \left\|\lambda_i^{-1} \boldsymbol{\delta}^{\top} \nabla_{\mathbf{x}} \mathbf{\Theta}(\mathbf{x}, \mathbf{X}) \mathbf{v_i} \mathbf{v_i}^{\top} \mathbf{Y} + \lambda_i^{-1} \mathbf{R} \mathbf{v_i} \mathbf{v_i}^{\top} \mathbf{Y}\right\|_2 && \text{(Where } \mathbf{R} : \text{remainder)} \\
&\overset{(b)}{\leq} \left\|\lambda_i^{-1} \boldsymbol{\delta}^{\top} \nabla_{\mathbf{x}} \mathbf{\Theta}(\mathbf{x}, \mathbf{X}) \mathbf{v_i} \mathbf{v_i}^{\top} \mathbf{Y} + \frac{\lambda_i^{-1} L}{2} \|\boldsymbol{\delta}\|_2 \mathbf{v_i} \mathbf{v_i}^{\top} \mathbf{Y}\right\|_2 && \text{(from (7))} \\
&\leq \lambda_i^{-1} \left\|\left(\boldsymbol{\delta}^{\top} \nabla_{\mathbf{x}} \mathbf{\Theta}(\mathbf{x}, \mathbf{X}) + \frac{L}{2} \|\boldsymbol{\delta}\|_2\right) \mathbf{v_i} \mathbf{v_i}^{\top} \mathbf{Y}\right\|_2 \\
&= \Theta\left(\frac{1}{\lambda_i}\right)
\end{aligned}
$$

where $(a)$ follows from the Taylor's expansion of $f_i^{ker}(\mathbf{x} + \boldsymbol{\delta})$ where $\mathbf{R}$ is the remainder terms and $(b)$ follows from (7), i.e., $\mathbf{R} \leq (L/2)\|\boldsymbol{\delta}\|_2$. $\qquad\square$

**Empirical validation: Top NTK features are indeed robust.** To verify Proposition 5.2, we construct a sanity experiment using a simple 1-layer NN $f(\mathbf{x}) = \frac{1}{d}\mathbf{w}^T \mathbf{x}$ with parameters $\mathbf{w} \in \mathbb{R}^d$ initialized from $\mathcal{N}(\mathbf{0}, \mathbf{I}_d)$. Let the data dimension $d$ be 100, the number of training samples $n$ be 1000 and the data is sampled from a Gaussian $\mathcal{N}(\mathbf{0}, \Sigma)$ where the covariance $\Sigma$ is a diagonal spiked matrix, that is, $\Sigma_{11} := 1 + \sqrt{d/n}$ and $\Sigma_{ii} := 1 \,\forall i \neq 1$. We then construct NTK features from the spectral decomposition of the exact NTK. Plot 3 of Figure 3 shows the norm of difference in the original, and adversarially perturbed NTK features with respect to the eigenvalues of the NTK spectrum for different perturbation strengths of $\Delta = \{0.01, 0.05, 0.1\}$. This validates our theory that the NTK features corresponding to the large eigenvalues are more robust and hence remain closer to the original feature even when perturbed.

## B Experiments

### B.1 Parameters for different algorithms

We set the parameters to the standard values in the literature. Refer to RobustBench for most of the attack parameters.

1. PGD: We perform PGD with the standard parameters in Table 8 to have an overall high strength PGD attack.

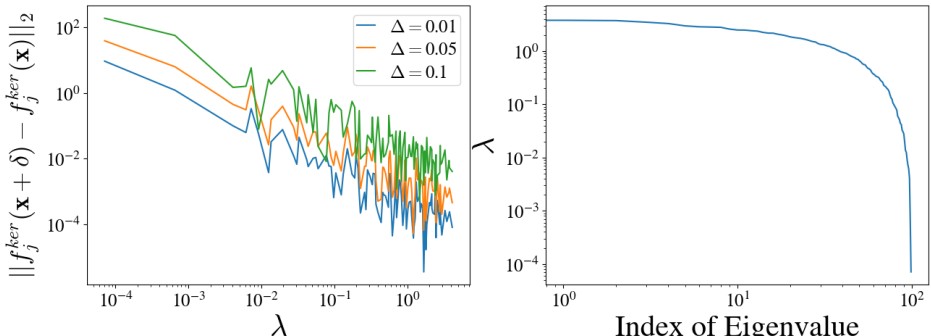

Figure 3: NTK feature robustness for $\lambda$ and the corresponding eigenvalue profile in ascending order.

Table 8: **Parameters for PGD.** We use these parameters for both training and attack.

| Dataset | $\ell_p$ | $\epsilon$ | step size | iteration |
|---|---|---|---|---|
| CIFAR-10, CIFAR-100 | $\ell_\infty$ | 8/255 | $\epsilon/4$ | 40 |
| | $\ell_2$ | 0.5 | $\epsilon/5$ | 100 |
| tiny ImageNet | $\ell_\infty$ | 4/255 | $\epsilon/4$ | 40 |

2. APGD-CE, APGD-DLR: We perform standard $\ell_\infty$ perturbation with the budget $\epsilon = 8/255$.

3. For Adversarial training in Table 10 we use same parameters for PGD, IAT, CW and TRADES as used in PGD attack from Table 8

## B.2 Details of benchmarking baseline methods

We perform benchmarking of our test-time defense on multiple SOTA methods that achieves adversarial robustness in the model. For our analysis of RFI on CIFAR-10 in table 10 we used PGD Madry et al. (2018), Interpolated Adversarial Training Lamb et al. (2019), Carlini-Warger Loss Carlini & Wagner (2017) and TRADES Zhang et al. (2019) to adversarially train the baseline model. In the case of Robust CIFAR-10 Ilyas et al. (2019), we only replaced the standard CIFAR-10 dataset with the robust dataset. In general for PGD, IAT and C&W attacks the adversarial training works as generating an adversarial example using the underlying attack and the objective is to minimize loss on these adversarial examples. PGD attack uses gradient descent to iteratively maximize the classification loss with respect to the input while projecting the perturbed example into the norm ball defined for the attack. IAT uses a joint objective that minimizes the classification loss of perturbed examples generated from PGD or any other attack along with classification loss on clean data with MixUP Zhang et al. (2017). We use Robust CIFAR-10 proposed in Ilyas et al. (2019), although is not an adversarial training method but rather the final dataset from a procedure to only retain robust features in the dataset. Ilyas et al. (2019) disentangle the robust and non-robust features by creating a one-to-one mapping of an input to its robustified image. From an adversarially pretrained backbone (ResNet-18 using PGD $\ell_2-$norm and $\epsilon = 0.25$) linear layer features are extracted for the natural image and also from a noise input. Then by minimizing the distance between these two representations in the input space over the noise, an image that only retains robust features of the original input is obtained.

For training using all these baseline adversarial training methods, we set the batch size as 128. We use SGD with momentum as the optimizer where we set the momentum to 0.1, we also set the weight decay to 0.0002. We run our training for 200 epochs and set the learning rate schedule as starting with 0.1 for the first 100 epochs and then we reduce the learning rate by 90 percent every 50 epochs. For calibration using temperature scaling (Guo et al., 2017), we take the trained model and optimize for the temperature parameter. The standard deviation in all the cases of calibrated models is reported by loading the pretrained models and 5 runs of calibration. Hence, there is no standard deviation for the non-calibrated models, and we also do not report the standard deviation for the SoTA models directly loaded from RobustBench.

### B.3 Adaptive attack performance of RFI on Expectation Over Transformation (EOT) attack using ResNet-18 for CIFAR-10

Expectation Over Transformation (EOT) is a procedure to synthesize examples that are adversarial over a chosen distribution of transformations (Athalye et al., 2018b). This procedure is shown to generate adversarial examples that are more robust to noise, distortions and affine transformations, and are consistently adversarial to the neural networks. EOT as an adversarial attack is observed to be stronger (Tramer et al., 2020) where a randomized transformation is applied to an input $\mathbf{x}$ before being fed into a classifier. RFI can be easily integrated into the neural network classifier in such settings by computing the transformation matrix $\tilde{\mathbf{U}}$ in RFI by applying random transformations to the training samples to ensure a similar distribution of the train and test sets.

We evaluate ResNet-18 using all the training settings considered in Table 1 on CIFAR-10 for Expectation Over Transformation (EOT) as an adaptive attack. The hyperparameters are the same as considered for adaptive attack evaluation in Section 4.1. We evaluate $\ell_\infty$ and $\ell_2$ attacks with $\epsilon = 8/255$ budget, $\epsilon/4$ step size and 40 iterations, and 0.5 budget, $\epsilon/5$ step size and 100 iterations, respectively. For RFI, we set $K = 10$. We observe that *RFI improves the performance by* 1 *to* 2% *consistently for EOT attack as well*.

Table 9: **Adaptive attack performance of RFI on Expectation over Transformation (EOT) attack**. We consider $\ell_\infty$ (step size $\epsilon/4$, 40 iterations) and $\ell_2$ (step size $\epsilon/5$, 100 iterations) attack on CIFAR-10 with ResNet-18. RFI improves robustness by 1 to 2% as shown in % Gain column.

| Training | Clean | | | $\ell_\infty(\epsilon = \frac{8}{255})$ | | | $\ell_2(\epsilon = 0.5)$ | | |
|---|---|---|---|---|---|---|---|---|---|
| | Method | +RFI | % Gain | Method | +RFI | % Gain | Method | +RFI | % Gain |
| PGD | $81.08_{\pm 0.01}$ | $80.49_{\pm 0.08}$ | **-0.59** | $36.01_{\pm 0.01}$ | $\mathbf{37.85}_{\pm 0.02}$ | **+1.84** | $35.52_{\pm 0.01}$ | $\mathbf{36.65}_{\pm 0.01}$ | **+1.13** |
| IAT | $90.32_{\pm 0.01}$ | $89.89_{\pm 0.01}$ | **-0.43** | $26.92_{\pm 0.00}$ | $\mathbf{28.30}_{\pm 0.01}$ | **+1.38** | $30.30_{\pm 0.00}$ | $\mathbf{31.47}_{\pm 0.02}$ | **+1.17** |
| C&W | $77.55_{\pm 0.03}$ | $77.50_{\pm 0.02}$ | **-0.05** | $22.51_{\pm 0.02}$ | $\mathbf{23.88}_{\pm 0.03}$ | **+1.37** | $25.71_{\pm 0.01}$ | $\mathbf{26.95}_{\pm 0.03}$ | **+1.24** |
| TRADES | $79.17_{\pm 0.02}$ | $79.02_{\pm 0.01}$ | **-0.15** | $47.20_{\pm 0.01}$ | $\mathbf{47.98}_{\pm 0.01}$ | **+0.78** | $48.55_{\pm 0.02}$ | $\mathbf{49.61}_{\pm 0.01}$ | **+1.06** |

### B.4 Adaptive attack performance of RFI on calibrated ResNet-18 for CIFAR-10

We evaluate ResNet-18 using all the training settings considered in Table 1 on CIFAR-10 for calibrated models. The hyperparameters are the same as non-calibrated setting. We evaluate $\ell_\infty$ and $\ell_2$ attacks with $\epsilon = 8/255$ budget, $\epsilon/4$ step size and 40 iterations, and 0.5 budget, $\epsilon/5$ step size and 100 iterations, respectively. For RFI, we set $K = 10$. We observe that *RFI improves the performance by* 4 *to* 9% *for calibrated models*.

Table 10: **Adaptive attack performance of RFI on calibrated models** using temperature scaling. We consider $\ell_\infty$ (step size $\epsilon/4$, 40 iterations) and $\ell_2$ (step size $\epsilon/5$, 100 iterations) attack on CIFAR-10 with ResNet-18. RFI improves robustness by 4 to 9% as shown in % Gain column.

| Training | Clean | | | $\ell_\infty(\epsilon = \frac{8}{255})$ | | | $\ell_2(\epsilon = 0.5)$ | | |
|---|---|---|---|---|---|---|---|---|---|
| | Method | +RFI | % Gain | Method | +RFI | % Gain | Method | +RFI | % Gain |
| Standard | $\mathbf{95.20}_{\pm 0.08}$ | $88.20_{\pm 0.10}$ | **-7.00** | $2.01_{\pm 0.38}$ | $\mathbf{6.83}_{\pm 0.22}$ | **+4.82** | $2.58_{\pm 0.62}$ | $\mathbf{10.21}_{\pm 0.81}$ | **+7.63** |
| Robust CIFAR-10 | $78.70_{\pm 0.04}$ | $\mathbf{78.73}_{\pm 0.06}$ | **+0.03** | $3.81_{\pm 0.14}$ | $\mathbf{8.03}_{\pm 0.21}$ | **+4.22** | $9.10_{\pm 0.92}$ | $\mathbf{11.21}_{\pm 0.68}$ | **+2.11** |
| PGD | $\mathbf{83.11}_{\pm 0.02}$ | $82.32_{\pm 0.08}$ | **-0.79** | $42.96_{\pm 0.75}$ | $\mathbf{50.08}_{\pm 0.88}$ | **+7.12** | $56.48_{\pm 0.42}$ | $\mathbf{62.13}_{\pm 0.92}$ | **+5.65** |
| IAT | $\mathbf{91.24}_{\pm 0.10}$ | $90.83_{\pm 0.08}$ | **-0.41** | $46.22_{\pm 0.10}$ | $\mathbf{51.34}_{\pm 0.83}$ | **+5.12** | $63.48_{\pm 0.96}$ | $\mathbf{71.12}_{\pm 0.29}$ | **+7.64** |
| C&W | $\mathbf{84.36}_{\pm 0.10}$ | $83.32_{\pm 0.05}$ | **-1.03** | $41.62_{\pm 0.90}$ | $\mathbf{50.48}_{\pm 1.07}$ | **+8.86** | $56.63_{\pm 0.68}$ | $\mathbf{63.21}_{\pm 0.72}$ | **+6.58** |
| TRADES | $\mathbf{81.11}_{\pm 0.01}$ | $79.38_{\pm 0.04}$ | **-1.73** | $53.67_{\pm 0.43}$ | $\mathbf{58.20}_{\pm 0.61}$ | **+4.53** | $62.12_{\pm 0.28}$ | $\mathbf{68.47}_{\pm 0.32}$ | **+6.35** |

### B.5 Adaptive attack performance of RFI for CIFAR-100 and tiny ImageNet

We evaluate both calibrated and non-calibrated ResNet-18 using all the adversarial training setting considered in Table 10 on CIFAR-100 since standard training would not result in robust model. We also consider tiny ImageNet dataset that has $100,000$ training and $10,000$ validation samples with 200 classes and ResNet-50 pretrained adversarially on ImageNet. We evaluate $\ell_\infty$ attack with $\epsilon = 8/255$ and $\epsilon = 4/255$ for CIFAR-100

and tiny ImageNet, respectively. The attack budget is standard, taken from RobustBench. For RFI, we set $K = 100$ and 200 (number of classes) for CIFAR-100 (Table 11) and tiny ImageNet (Table 12), respectively. % Gain in tables is between Calibration+RFI and the base method.

Table 11: **Adaptive attack performance of RFI on non-calibrated and calibrated models.** Robust performance evaluation of RFI on CIFAR-100 with ResNet-18 (step size $\epsilon/4$ and 40 iterations). RFI improves the performace on an average by **4**%.

| Training | Clean | | | | | $\ell_\infty(\epsilon = \frac{8}{255})$ | | | | |
|---|---|---|---|---|---|---|---|---|---|---|
| | Method | +RFI | +Calibration | +Calibration+RFI | % Gain | Method | +RFI | +Calibration | +Calibration+RFI | % Gain |
| PGD | 55.30 | 55.27 | **55.82** | 55.08 | **-0.22** | 20.08 | 20.91 | 21.86 | **25.96** | **+5.88** |
| IAT | **58.94** | 58.88 | 58.86 | 58.09 | **-0.85** | 22.56 | 23.58 | 23.04 | **26.72** | **+4.16** |
| C&W | **49.36** | 49.31 | 49.30 | 49.02 | **-0.34** | 10.44 | 11.86 | 11.28 | **14.72** | **+4.28** |
| TRADES | **55.17** | 55.11 | **55.17** | 55.10 | **-0.07** | 28.25 | 28.56 | 28.43 | **30.91** | **+2.66** |

In the case of tiny ImageNet, we subsampled 100 training samples per class instead of using the full training set for computing the transformation matrix $\tilde{U}$ of the feature covariance due to the computation time, and evaluated the clean and robust performances on the $10,000$ validation samples. The results are given in Tables 11 and 12. We observe that *RFI consistently improves the adversarial performance on the datasets with a very small drop in the clean performance.* Thus this shows RFI generalizes to larger datasets as well. Furthermore, we would like to draw the attention that *our method improves the performance even with a small subsample of the dataset.*

Table 12: **Adaptive attack performance of RFI on non-calibrated and calibrated models.** Robust performance evaluation of RFI on tiny ImageNet with ResNet-50 (step size $\epsilon/4$ and 40 iterations). RFI improves robustness even on large datasets.

| Training | Clean | | | | | $\ell_\infty(\epsilon = \frac{4}{255})$ | | | | |
|---|---|---|---|---|---|---|---|---|---|---|
| | Method | +RFI | +Calibration | +Calibration+RFI | % Gain | Method | +RFI | +Calibration | +Calibration+RFI | % Gain |
| PGD | **62.42** | 62.39 | 62.40 | 62.32 | **-0.10** | 33.38 | **33.50** | 33.43 | 34.27 | **+0.89** |

## B.6 Adaptive attack performance of RFI on state-of-the-art models from RobustBench

For table 4 we benchmark our test-time defense on multiple recent SoTA methods for CIFAR-10, CIFAR-100 and ImageNet. For all our baseline methods we obtain the model weights from RobustBench Croce & Hein (2020b). We update the weights of the last linear layer of the models using RFI and benchmark the updated models. We also report the performance for optimal $K$ in RFI. We note that the Expected Calibration Error (ECE) for the SoTA models are very small as shown in Table 14 (already well calibrated), hence we do not explicitly calibrate in Table 13. Moreover, the results in Table 4 show that calibration will only further improve robustness with RFI. Therefore, we do conservative analysis of RFI on the SoTA models. For Salman et al. (2020) on ImageNet we compute with and without dynamic RFI and not Anti-Adv and SODEF since it increase the inference costs of the evaluation such that we could no longer run experiments with our computational resources. Also we do not report AutoAttack since it requires all 4 attacks i.e. APGD-CE, APGD-DLR, FAB and Square to be executed sequentially which is outside the scope of max runtime of our resources. Nevertheless, we observe that *RFI improves the robustness reliably $\sim 1.5\%$ on average on non-calibrated SoTA models.* Importantly, SODEF and Anti-adv reduces the robustness performance especially on AutoAttack which is inline to the findings of Croce & Hein (2020b).

## B.7 Transferability Study

We conduct a more detailed transferability of attack analysis on CIFAR-10 using ResNet-18 and on CIFAR-100 using PreActResNet-18. Here, we generated adversarial examples with respect to the base model and all the defences and evaluated the robustness of different adaptive defences under all the adversary cases (Transfer attacks). Then we present the results for calibrated ResNet-18 on CIFAR-10 in Table 16 which completes the analysis together with the results from Table 2. We observe that the robustness of the calibrated model

Table 13: **Adaptive attack performance evaluation of RFI on state-of-the-art methods.** We apply APGD-CE, APGD-DLR and RobustBench attacks on CIFAR-10 and CIFAR-100. The inference time for RFI is $1\times$, whereas Anti-adv and SODEF are $8\times$ and $2\times$, respectively. There is no standard deviation as the trained models are directly from RobustBench. While RFI improves the robustness to AutoAttack **upto 1.5%** without calibration, SODEF and Anti-adv results in **no ($< 0.1\%$)** or **decrease** in robustness consistently.

| | Base Method | Defense | Clean | APGD-CE | APGD-DLR | FAB | Square | AutoAttack |
|---|---|---|---|---|---|---|---|---|
| **CIFAR-10** | Carmon et al. (2019) WideResNet-28-10 | None | **89.69** | 61.82 | 60.85 | 60.18 | 66.51 | 59.53 |
| | | Anti-adv | **89.69** | 61.81 | 60.89 | 60.11 | 66.58 | 58.70 |
| | | SODEF | 89.68 | 60.20 | 60.72 | 58.04 | 65.28 | 57.23 |
| | | RFI ($K = 10$) | 89.60 | 62.38 | 61.58 | 60.21 | 66.59 | 60.72 |
| | | RFI (opt. $K = 20$) | 89.60 | **62.45** | **61.60** | **60.38** | **66.90** | **61.02** |
| | Engstrom et al. (2019) ResNet-50 | None | **87.03** | 51.75 | 60.10 | 49.90 | 58.00 | 49.25 |
| | | Anti-adv | 87.00 | 51.62 | 59.95 | 49.84 | 58.06 | 49.20 |
| | | SODEF | 86.95 | 50.01 | 58.20 | 48.64 | 56.68 | 47.92 |
| | | RFI ($K = 10$) | 87.01 | 51.86 | 61.84 | 51.28 | 58.07 | 50.75 |
| | | RFI (opt. $K = 15$) | **87.03** | **51.94** | **61.90** | **51.46** | **58.12** | **50.98** |
| | Rice et al. (2020) WideResNet-34-10 | None | 85.34 | 50.12 | 56.80 | 53.87 | 56.88 | 53.42 |
| | | Anti-adv | **85.40** | 50.10 | 57.50 | 53.90 | 57.00 | 50.98 |
| | | SODEF | 85.10 | 50.60 | 56.50 | 53.72 | 56.21 | 50.09 |
| | | RFI($K = 10$) | 85.30 | 51.19 | 58.55 | 53.98 | **57.13** | 54.64 |
| | | RFI (opt. $K = 35$) | 85.30 | **51.62** | **58.97** | **54.12** | **57.13** | **54.86** |
| | Wang et al. (2023) WideResNet-28-10 | None | **92.44** | 70.23 | 67.82 | 67.41 | 73.13 | 67.31 |
| | | Anti-adv | **92.44** | 68.90 | 65.91 | 67.55 | 73.20 | 66.52 |
| | | SODEF | 92.01 | 67.53 | 65.08 | 65.93 | 73.01 | 64.20 |
| | | RFI ($K = 10$) | 92.33 | 70.32 | 67.86 | **67.82** | 73.52 | 67.29 |
| | | RFI (opt. $K = 20$) | 92.34 | **70.36** | **67.90** | **67.82** | **73.54** | **67.50** |
| **CIFAR-100** | Pang et al. (2022) WideResNet-28-10 | None | **63.66** | 35.29 | 31.71 | 31.32 | 35.70 | 31.08 |
| | | Anti-adv | 63.41 | 32.50 | 30.32 | 31.30 | 35.76 | 30.10 |
| | | SODEF | 63.08 | 30.96 | 29.54 | 31.44 | 32.27 | 30.56 |
| | | RFI ($K = 100$) | 63.01 | 36.03 | **31.95** | 31.88 | 35.79 | 31.29 |
| | | RFI (opt. $K = 115$) | 63.10 | **36.07** | **31.95** | **31.96** | **35.88** | **31.91** |
| | Addepalli et al. (2022) ResNet-18 | None | **65.45** | 33.49 | 28.55 | 28.00 | 33.70 | 27.67 |
| | | Anti-adv | 65.38 | 30.92 | 26.61 | 27.92 | 33.61 | 26.01 |
| | | SODEF | 65.23 | 29.37 | 26.90 | 24.62 | 29.60 | 26.53 |
| | | RFI ($K =$ opt. $K = 100$) | 65.41 | **34.09** | **29.18** | **28.10** | **33.79** | **27.80** |
| | Rice et al. (2020) PreActResNet-18 | None | **53.83** | 20.83 | 20.46 | **23.82** | 19.29 | 18.95 |
| | | Anti-adv | **53.83** | 20.78 | 20.06 | 23.49 | 19.27 | 18.97 |
| | | SODEF | **53.83** | 18.50 | 19.20 | 19.66 | 16.05 | 16.92 |
| | | RFI ($K = 100$) | 53.70 | 21.10 | 20.98 | 20.93 | 18.13 | 19.23 |
| | | RFI (opt. $K = 150$) | 53.75 | **21.18** | **21.10** | 21.03 | **19.53** | **19.46** |
| | Wang et al. (2023) WideResNet-28-10 | None | **72.58** | 44.04 | 39.78 | 39.19 | 44.46 | 38.83 |
| | | Anti-adv | 72.57 | 42.98 | 38.10 | 36.85 | 44.49 | 34.01 |
| | | SODEF | 72.34 | 38.10 | 36.95 | 34.82 | 44.42 | 32.29 |
| | | RFI ($K = 100$) | 72.55 | 44.37 | 39.91 | 39.68 | 44.50 | 39.10 |
| | | RFI (opt. $K = 115$) | 72.55 | **44.51** | **39.96** | **39.81** | **44.53** | **39.13** |
| **ImageNet** | Salman et al. (2020) ResNet-50 | None | **64.02** | 38.32 | 34.02 | 34.35 | 49.52 | - |
| | | Dynamic RFI | 63.91 | **38.48** | **34.68** | **34.68** | **49.98** | - |
| | Salman et al. (2020) WideResNet-50-2 | None | **68.46** | 40.67 | 37.09 | 37.81 | 54.61 | - |
| | | Dynamic RFI | 68.41 | **40.84** | **37.56** | **38.12** | **54.78** | - |

with RFI is on par with the base calibrated model. Moreover, when attacked with examples from RFI integrated model, the base model performs worse. Notably, *the decrease in robust performance of the base method is much more than the decrease of the performance of RFI when evaluated on adversary from the base method.* This shows RFI's goodness and further confirms the absence of an obfuscated gradient in RFI. Similar observation using a SoTA model on CIFAR-100 are in Table 15 (Appendix). In contrast, transfer

Table 14: **Expected Calibration Error (ECE) of the SoTA models** are very small, hence already well calibrated.

| CIFAR-10 | | | | CIFAR-100 | | |
|---|---|---|---|---|---|---|
| Method | ECE | ECE after Calibration | | Method | ECE | ECE after Calibration |
| Carmon et al. (2019) WideResNet-28-10 | 4.310 | **0.328** | | Pang et al. (2022) WideResNet-28-10 | 0.364 | **0.142** |
| Engstrom et al. (2019) ResNet-50 | 0.091 | **0.065** | | Addepalli et al. (2022) ResNet-18 | 0.418 | **0.347** |
| Rice et al. (2020) WideResNet-34-10 | 0.074 | **0.037** | | Rice et al. (2020) PreActResNet-18 | 0.138 | **0.074** |
| Wang et al. (2023) WideResNet-28-10 | 0.145 | **0.039** | | Wang et al. (2023) WideResNet-28-10 | 0.366 | **0.290** |

attacks from base model on SODEF show a significant drop in robustness (Section 4.6.2 of Kang et al. (2021)) and on Anti-adv render the defense ineffective (Section 3.8 of Croce et al. (2022)). These results further highlight the soundness of RFI.

### B.7.1 RFI results in stronger adversary against transfer from other defenses

In the set of experiments, we evaluate all combinations of transfer attacks on CIFAR-100 and PreActResNet-18 Rice et al. (2020) in Table 15. We compare the transferability of all adaptive test-time defenses to base model and within themselves by using adversarial examples generated with one defense attacking another defense. The general observation and expectation is that the model performance is affected the most when the adversarial examples are created using the same model, i.e., adaptive attack. This observation holds in our experiments too. The most interesting and impressive observation is that *RFI outperforms all other methods in almost all the cases, even when adversarial examples are generated from base model + RFI.* Notice that SODEF and Anti-adv suffer the most when adversarial examples are generated from the respective models, unlike RFI showing the impressive robustness of our method.

Table 15: **Transfer attack on non-calibrated PreActResNet-18 for CIFAR-100.** RFI outperforms in all the cases and also generates the strongest adversary for the base model.

| Adversarial Examples are generated from **Method (Rice et al)** | | | | |
|---|---|---|---|---|
| Attack | Method | +AntiAdv | +SODEF | +RFI |
| APGD-CE | 20.83 | 20.06 | 27.13 | **27.30** |
| APGD-DLR | 20.46 | 20.52 | 29.33 | **29.53** |
| FAB | 19.29 | 19.28 | 35.38 | **35.90** |
| Square | 23.82 | 23.58 | 36.83 | **36.88** |
| AutoAttack | 18.95 | 18.97 | 26.09 | **26.43** |

| Adversarial Examples are generated from **Method+AntiAdv** | | | | |
|---|---|---|---|---|
| Attack | Method | +AntiAdv | +SODEF | +RFI |
| APGD-CE | 20.59 | 20.58 | **27.31** | 26.65 |
| APGD-DLR | 20.39 | 20.49 | **28.92** | 28.53 |
| FAB | 19.27 | 19.27 | 35.80 | **38.69** |
| Square | 23.60 | 23.49 | 37.41 | **39.04** |
| AutoAttack | 18.98 | 18.96 | 25.61 | **26.15** |

| Adversarial Examples are generated from **Method+SODEF** | | | | |
|---|---|---|---|---|
| Attack | Method | +AntiAdv | +SODEF | +RFI |
| APGD-CE | 32.99 | **37.32** | 18.50 | 37.30 |
| APGD-DLR | 33.65 | **38.34** | 19.20 | 38.30 |
| FAB | 39.67 | 48.11 | 16.05 | **48.12** |
| Square | 39.59 | 48.20 | 19.66 | **48.22** |
| AutoAttack | 32.76 | 33.16 | 15.69 | **37.23** |

| Adversarial Examples are generated from **Method+RFI** | | | | |
|---|---|---|---|---|
| Attack | Method | +AntiAdv | +SODEF | +RFI |
| APGD-CE | 14.70 | 18.31 | 18.40 | **21.18** |
| APGD-DLR | 14.12 | 18.30 | 19.21 | **21.10** |
| FAB | 12.76 | 14.12 | 14.70 | **18.13** |
| Square | 16.29 | 18.95 | 19.50 | **20.93** |
| AutoAttack | 12.55 | 16.50 | 16.92 | **19.46** |

### B.8 Transfer attack: RFI with calibration is on par with the base model

Results on transfer attacks, where we assess the performance of RFI against adversarial samples generated from the base, for calibrated and on CIFAR-10 with Resnet 18 backbone are in Tables 16. Notably, *RFI demonstrates comparable robustness to the base model*, ensuring that gradient obfuscation is *not* at play in RFI and affirming that it reliably improves the model robustness. Moreover, the transferability of adversary from RFI leads to a degradation in robustness for the base model, suggesting that *RFI acts as an on-par adversary to the base* (refer to +RFI rows of the left subtable in the Table 16). We hypothesize that the attack from base model and attack from base model + RFI affect different semantics or examples such that

on average both are on par post-calibration. As expected the adversarial samples from the base method + RFI are more powerful and reduce the robustness of the base method to a greater extent than vice versa.

Table 16: **Transfer attack performance of RFI on calibrated models.** RFI is on par with the base, ensuring reliable robustness improvement without gradient obfuscation. Setting same as Table 10. The decrease in robustness of the base model is much more than the robustness of RFI when evaluated on the adversary from the base.

| | Adversary generated from **base model+RFI** | | | |
|---|---|---|---|---|
| Training | $\ell_\infty(\epsilon = \frac{8}{255})$ | | $\ell_2(\epsilon = 0.5)$ | |
| | Method | +RFI | Method | +RFI |
| PGD | $42.85 \pm 0.12$ | $\mathbf{50.08} \pm 0.88$ | $57.18 \pm 0.54$ | $\mathbf{62.13} \pm 0.92$ |
| IAT | $47.92 \pm 0.31$ | $\mathbf{51.34} \pm 0.83$ | $64.38 \pm 0.33$ | $\mathbf{71.12} \pm 0.29$ |
| C&W | $40.73 \pm 0.64$ | $\mathbf{50.48} \pm 1.07$ | $55.96 \pm 0.88$ | $\mathbf{63.21} \pm 0.72$ |
| TRADES | $55.43 \pm 0.42$ | $\mathbf{58.20} \pm 0.61$ | $64.34 \pm 0.40$ | $\mathbf{68.47} \pm 0.32$ |
| | Adversary generated from **base method**. | | | |
| Training | $\ell_\infty(\epsilon = \frac{8}{255})$ | | $\ell_2(\epsilon = 0.5)$ | |
| | Method | +RFI | Method | +RFI |
| PGD | $\mathbf{42.96} \pm 0.75$ | $41.38 \pm 0.48$ | $\mathbf{56.48} \pm 0.42$ | $54.28 \pm 0.62$ |
| IAT | $\mathbf{46.22} \pm 0.10$ | $43.44 \pm 0.21$ | $\mathbf{63.48} \pm 0.96$ | $62.19 \pm 0.09$ |
| C&W | $\mathbf{41.62} \pm 0.90$ | $39.10 \pm 0.81$ | $\mathbf{56.63} \pm 0.68$ | $55.19 \pm 0.91$ |
| TRADES | $\mathbf{53.67} \pm 0.43$ | $52.88 \pm 0.33$ | $\mathbf{62.12} \pm 0.28$ | $59.85 \pm 0.97$ |

### B.9 Static vs Dynamic RFI on calibrated model

We extend the study of static vs dynamic RFI to calibrated models in this section using the same setup as Section 4.3, where we consider pretrained ResNet-18 on CIFAR-10 by applying PGD ($\ell_\infty, \epsilon = 8/255$) and ($\ell_2, \epsilon = 0.5$) in transfer attack setting i.e. generate adversarial examples from the base method. For the dynamic setting we compute the covariance batch-wise to compute $\tilde{U}$ with the input. Table 17 shows *static is better than dynamic RFI similar to non-calibrated setting.*

Table 17: **Additional Comparison of static and dynamic/adaptive RFI on calibrated model showing static RFI is better than dynamic RFI.** Setting same as Table 10. Adversarial examples are generated from the base model for fair comparison.

| Training | Clean | | $\ell_\infty(\epsilon = \frac{8}{255})$ | | $\ell_2(\epsilon = 0.5)$ | |
|---|---|---|---|---|---|---|
| | Static | Dynamic | Static | Dynamic | Static | Dynamic |
| Standard | 10.36 | **11.65** | 20.08 | 11.64 | **20.91** | 12.43 |
| Robust CIFAR-10 | **78.78** | 75.23 | **15.41** | 12.89 | **17.38** | 16.32 |
| PGD | 83.22 | 82.86 | 46.02 | **46.83** | 58.81 | **59.23** |
| IAT | 91.26 | **91.35** | **49.06** | 48.53 | **66.67** | 66.28 |
| C&W | 84.97 | 83.01 | **45.48** | 43.98 | **58.95** | 57.82 |
| TRADES | 80.76 | 78.98 | **54.33** | 53.58 | **65.23** | 65.00 |

### B.10 Static RFI is Optimal

In the case of dynamic RFI implementation, one needs to know when to apply the transformation as the model should be static for the attacker and adapted only for the defender. This poses implementation difficulty as the situation is mostly unknown in practice. Hence, we explore different variants of RFI in a dynamic setting where we compute the covariance matrix and eventually the transformation matrix $\tilde{U}$ using the full validation set or single test input. We observe that *the dynamic RFI is only marginally better than the static RFI* when full validation set is used in Table 18 (a). Similarly, we present the result for single test

input in Table 18 where *the method shows improvement in clean performance* since it is only a normalization of the feature representation. We perform these comparisons to highlight the fact that these hypothetical variants of dynamic RFI which work with information of validation set are also not significantly better than static RFI, thereby implying that **static RFI is indeed the optimal way of selecting $\tilde{U}$ as indicated by our theory**.

Table 18: **Static RFI is the optimal approach.** RFI with covariance matrix calculated using different approaches.

(a) RFI with covariance matrix calculated using complete validation set

| Training | Clean | | $\ell_\infty(\epsilon = \frac{8}{255})$ | | $\ell_2(\epsilon = 0.5)$ | |
|---|---|---|---|---|---|---|
| | Method | +RFI | Method | +RFI | Method | +RFI |
| Standard | **95.28** | 88.53 | 1.02 | **9.35** | 0.39 | **11.73** |
| Robust CIFAR-10 | 78.69 | **78.80** | 1.30 | **11.21** | 9.63 | **12.56** |
| PGD | **83.53** | 83.29 | 42.20 | **43.82** | 54.61 | **56.13** |
| IAT | **91.86** | 91.32 | 44.76 | **47.65** | 62.53 | **64.88** |
| C&W | **85.11** | 85.06 | 40.01 | **43.48** | 55.02 | **57.83** |
| TRADES | **81.13** | 80.97 | 51.70 | **54.29** | 60.03 | **61.79** |

(b) RFI with covariance matrix calculated using single test input

| Training | Clean | | $\ell_\infty(\epsilon = \frac{8}{255})$ | | $\ell_2(\epsilon = 0.5)$ | |
|---|---|---|---|---|---|---|
| | Method | +RFI | Method | +RFI | Method | +RFI |
| Standard | **95.28** | 90.10 | 1.02 | **10.81** | 0.39 | **12.16** |
| Robust CIFAR-10 | 78.69 | **78.70** | 1.30 | **11.88** | 9.63 | **12.87** |
| PGD | **83.53** | 83.52 | 42.20 | **44.08** | 54.61 | **56.53** |
| IAT | **91.86** | 91.86 | 44.76 | **47.95** | 62.53 | **65.01** |
| C&W | **85.11** | 85.11 | 40.01 | **43.48** | 55.02 | **58.09** |
| TRADES | **81.13** | 81.09 | 51.70 | **54.78** | 60.03 | **62.17** |

## B.11 Ablation study

### B.11.1 Effect of $K$

Neural Collapse is a phenomenon in which the penultimate feature of each class collapses to its mean after the training error reaches zero. This implies that there is principally only $C = \#classes$ number of feature vectors, one for each class. Hence, we suggest setting $K$ to number of classes. We also justify it experimentally in Figure 2. Additional experiments for large-scale models used in Tables 4 and 13 with respect to CIFAR-10 and CIFAR-100 also show drop in eigenvalues at the number of classes across models, justifying our choice for $K$. We also extend the ablation study on $K$ to report the best performance in Figure 2 and the optimal $K$ row in 13. Note that the optimal $K$ for robust performance is not the best for standard performance as we are choosing only the top-most informative features (Corollary 3.6).

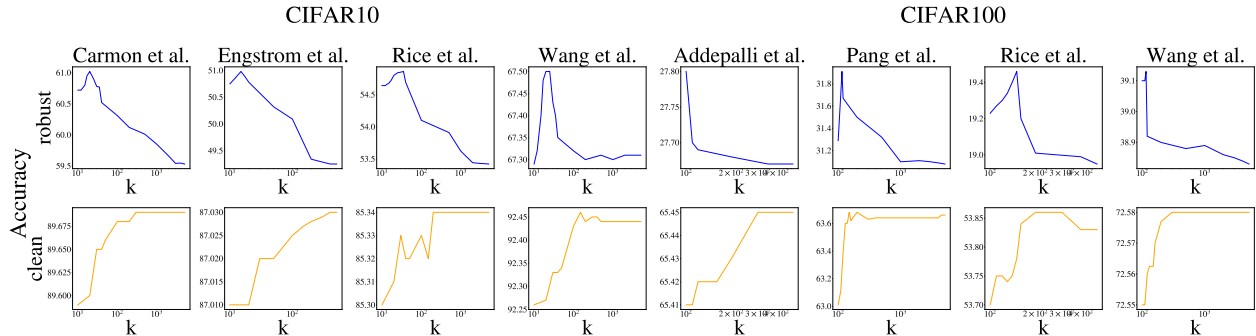

Figure 4: **Ablation of performance with $K$ for all SoTA models for CIFAR-10 and CIFAR-100.**

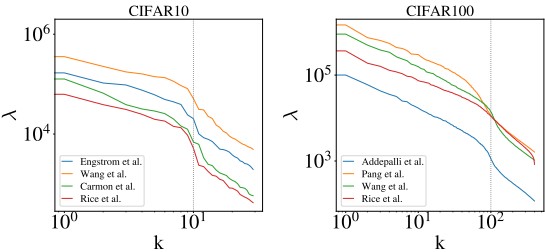

Figure 5: **Eigenspectrum showing sharp drop at $K$ = number of classes** for all SoTA models on CIFAR-10 and CIFAR-100.

### B.11.2 Effect of step size in PGD

We chose $\epsilon/4$ and $\epsilon/5$ for step sizes in $\ell_\infty$ and $\ell_2$, respectively, following the benchmarks in several works in RobustBench. The other common choice for the step size is proportional to the iterations, that is, $2\epsilon/40$ and $2\epsilon/100$ for $\ell_\infty$ and $\ell_2$, respectively. We reevaluated the models in Table 10 with and without RFI for these step sizes and the results are in Table 19, showing that *RFI is better than the base model*, in line with the observations in the previous experiments.

Table 19: **RFI is more robust than the base model irrespective of the step size in PGD.** $2\epsilon/40$ for $\ell_\infty$ and $2\epsilon/100$ for $\ell_2$.

| Training | $\ell_\infty(\epsilon = \frac{8}{255})$ | | $\ell_2(\epsilon = 0.5)$ | |
|---|---|---|---|---|
| | Method | +RFI | Method | +RFI |
| Standard | 0.03 | **9.73** | 3.67 | **14.13** |
| PGD | 44.44 | **45.48** | 57.77 | **58.97** |
| IAT | 45.91 | **48.26** | 66.26 | **67.73** |
| Robust CIFAR10 | 7.14 | **15.57** | 12.94 | **17.15** |
| CW Attack | 38.89 | **41.53** | 51.20 | **54.45** |
| TRADES | 52.90 | **54.10** | 61.66 | **63.35** |

### B.12 Conceptual ideas similar to RFI

**Low dimensional last layer.** Similar to comparing RFI on last layer vs on intermediate layer in Section 4.5.3, here we compare RFI and directly training a network with $K$ neurons in the last layer. We consider two ResNet-18 models with an additional fully connected hidden layer of size 512 and 10, respectively, and are trained with PGD. We apply RFI only to the larger model with 512 neurons and reduce the dimension to 10, and compare the performances in terms of clean and robust accuracies in both cases. The results are presented in Table 20, showing that *RFI is more robust compared to imposing a low dimensional last layer.*

Table 20: **RFI is more robust compared to imposing a low dimensional last layer.** ResNet-18 with last hidden layer size 10 and 512. RFI done on model with 512 hidden layer.

| | +hidden layer=10 | +hidden layer=512 | +hidden layer=512 + RFI |
|---|---|---|---|
| Clean | 83.71 | **84.13** | 84.05 |
| Robust ($\ell_\infty$) | 42.43 | 42.73 | **43.53** |

We further argue qualitatively why setting low dimension layers is not equivalent to RFI as follows. Firstly, overparameterization is shown empirically to be the key for both generalization Brutzkus & Globerson (2019) and robustness Madry et al. (2018). Especially in the case of CNN, there is an empirical understanding to build the network with more than one fully connected layer after the convolution layers starting with larger widths to generalize well Bengio (2012). These findings oppose the idea of having low dimension for the last hidden layer. Secondly, there are similar insights from the sparsity of neural networks – a smaller subnetwork with similar performance can be obtained by sparifying the network, called a lottery ticket Frankle & Carbin (2018). Once known, lottery tickets can be trained from scratch to reach similar performance as the original network. However, it is not possible to obtain the ticket simply by setting hyperparameters for a smaller network from the beginning. Finally, we emphasize that with RFI the last hidden-layer dimension is reduced by a large amount in comparison to the actual model. For example, in CIFAR-10, ResNet-50 with 2048 dimensions is reduced to $10(= K)$. So, the network with 10 dimension conventionally would not help generalization, which is conclusively established in the above experiment.

### B.13 Visualization of robust and non-robust features

We obtain the visualizations of robust and non-robust features for an input $\mathbf{x}$ by solving

$$\arg\min_{\tilde{\mathbf{x}}} ||\Phi(\tilde{\mathbf{x}}) - \Phi(\mathbf{x})\mathbf{U}\mathbf{U}^T||_2$$

where $\mathbf{U}$ is top $K$ eigenvectors based on $s_c(.)$ for robust features and all eigenvectors except top $K$ for non-robust features. The objective is solved using gradient descent. Figure 6 shows the visualizations of features for a few classes in CIFAR-10 using the PGD adversarially trained ResNet-18 model. 'Robust $K = 10$' and 'Non-robust $K = 10$' columns are obtained by setting $\mathbf{U}$ to the top $K$ eigenvectors and everything except the top $K$ eigenvectors based on $s_c(.)$, respectively. The columns top and bottom 100 eigenvectors are obtained by setting $\mathbf{U}$ to the top and bottom 100 eigenvectors based on the eigenvalues. The feature visualizations show that robust and top eigenvectors result in more similar features. The interesting observation is that the non-robust and bottom eigenvectors are equally noisy and might have some useful information that reflects the drop in clean performance. Nevertheless, it is not possible to argue based on the visual interpretation of the features since the difference is primarily coming from the eigenspace of the feature covariance.

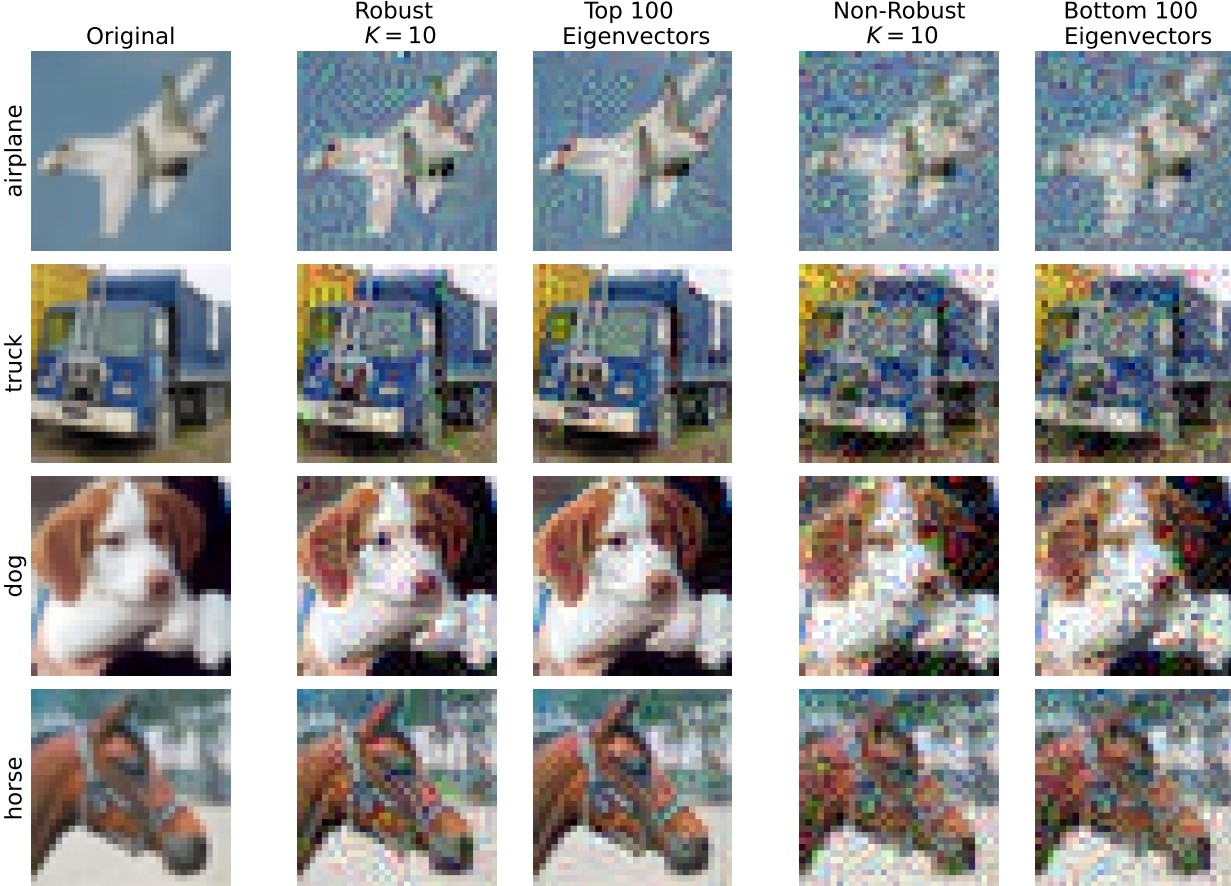

Figure 6: **Robust and non-robust features visualization**. The features are obtained using the PGD adversarially trained ResNet-18 model, and the original images are from CIFAR-10. The columns robust $K = 10$ are the robust features by fixing $\mathbf{U}$ to top $K$ eigenvectors based on the score function $s_c(.)$, whereas top 100 eigenvectors is based on the largest 100 eigenvalues. Likewise, non-robust $K = 10$ are obtained by fixing $\mathbf{U}$ to all eigenvectors except the ones in robust $K = 10$ and bottom 100 eigenvectors are obtained using the smallest 100 eigenvalues, respectively.

