# OpenReview forum: "Robust Feature Inference: A Test-time Defense Strategy using Spectral Projections"
_TMLR — Accepted by TMLR_

### Review · Reviewer_kcT8 · 2024-05-29

**Summary Of Contributions:**

The paper designs a robust feature inference (RFI) approach based on principal component analysis of the training features. The top-K principal components are selected as more robust features according to the theoretical analysis in Sec. 3.1. The proposed RFI is tested on five datasets using both white-box and black-box attacks and is demonstrated to improve the original model's adversarial robustness across different experimental settings.

**Audience:**

Yes

**Broader Impact Concerns:**

Not applicable.

**Claims And Evidence:**

Yes

**Requested Changes:**

1. It looks like the RFI is more effective when using the $l_{\infty}$-norm attack than the $l_2$ attack, which is a bit of weird to me as the theoretical part defines the robustness with respect to $l_2$ norm.

2. Please make Fig. 2 high-resolusion using the pdf/svg format.

**Strengths And Weaknesses:**

Strengths:

The paper is well written and the designed method is presented well. The empirical study is extensive and corroborates the effectiveness of the proposed RFI.


Weaknesses:

According to the guidelines of TMLR, the paper does not have obvious weaknesses as the empirical evidence is sufficient and strong.

If I have to say some weakness, the effectiveness of RFI is not quite strong on CIFAR100 when AutoAttack is used, according to Table 4..

---

> ### Author Response · Authors · 2024-06-16
>
> We thank the reviewer for the positive feedback on our work and address the requested changes below.
>
> ### RFI more effective for $\ell_\infty$ than $\ell_2$
> Even though the theory is based on feature definitions with respect to $\ell_2$, it does not directly yield a way to compare different $\ell_p$ norms, especially under different perturbation budgets $\epsilon$. Extending our theory to $\ell_\infty$ is not straightforward, and also, standardizing the comparison of performance gains in different $\ell_p$ norms is difficult. That being said, the ablation study on the effect of adversary strength in Section 4.5.1 shows that the trend in the performance gain between $\ell_\infty$ and $\ell_2$ as we change the budget $\epsilon$ is similar (Table 6).
>
> ### High-resolution image
> We have updated the figure in the draft with a high-resolution one.

---

### Review · Reviewer_Rc5R · 2024-06-02

**Summary Of Contributions:**

This paper introduces a novel test-time defense method designed to counter adversarial attacks during inference. Initially, it identifies the concept of robust features, subsequently developing a comprehensive theory and an algorithm to determine these features. Furthermore, leveraging these robust features, the authors have crafted efficient algorithms for classifying test data, resulting in enhanced robust performance. Experimental results back up the proposed algorithm.

**Audience:**

Yes

**Broader Impact Concerns:**

No concern

**Claims And Evidence:**

Yes

**Requested Changes:**

See the weakness section above.

**Strengths And Weaknesses:**

Strengths:

* This paper proposes a new test-time defense algorithm, which can be seamlessly integrated with any robust learning algorithms in the training process.

* This paper develops a theoretical analysis for the intuition and the developed algorithm, which justifies the validity of the proposed algorithm.

* Experimental results highlight the superior performance of the developed algorithm, improving baseline robust accuracy by 1%-2%.

Weaknesses:

* The study primarily focuses on the output layer of the neural network, leaving the hidden layers and their representations, i.e., the representation $\Phi(x)$  unchanged. However, in practice, $\Phi(x)$ cmay also be sensitive to perturbations for some inputs, it would be beneficial to consider the sensitivity of  $\Phi(x)$ as well.

* Additional visualizations or experimental discussions regarding the robust and unrobust features could enhance understanding. These terms are widely used in literature and might differ from their usage in this paper, thus a more detailed discussion should be added.

* The impact of the defense algorithm on clean accuracy is noted, but not theoretically justified. Including such justification could strengthen the paper's conclusions.

* Considering the test-time defense approach presented, exploring whether a corresponding adversarial attack method could be designed—specifically targeting the space of robust features with all available perturbation budget—would be an interesting extension of this work.

---

> ### Author Response · Authors · 2024-06-16
>
> We thank the reviewer for the constructive feedback and for pointing out the weaknesses in our initial submission. In the following, we describe how we address them in the updated manuscript.
>
> ### Sensitivity of $\Phi(x)$
> While it is theoretically challenging to analyze the sensitivity of features from intermediary layers due to the non-linearity and the complex training dynamics of the neural networks, we experimentally explored the idea of RFI to the intermediary layers by projecting the last but one layer in Section 4.6.1. Table 7 shows that the RFI on intermediary layers actually harms the robustness. Although this is not a direct sensitivity analysis of $\Phi(x)$, this experiment suggests that the theoretical result concerning the top subspace of the covariance of the output layer feature might not transfer to the intermediary layers.
>
> Additionally, [Zeiler and Fergus, 2014] showed that the features are learned hierarchically in neural networks. That is, the initial layers learn simpler features like edge/color detectors and textures, while the latter layers learn more complex and class-specific features. This suggests that the layers become progressively more sensitive, and hence, analyzing the robustness of the last layer should suffice.
>
>
> ### Additional visualizations of the robust and non-robust features
> We provide both the visualizations of robust and non-robust features in Appendix B.13, and a discussion on the difference to other feature definitions in the main draft.
> 1. **Visualization:** we obtain the visualizations of robust and non-robust features for an input $\mathbf{x}$ by solving $\arg\min_{\tilde{\mathbf{x}}} || \Phi(\tilde{\mathbf{x}}) - \Phi(\mathbf{x})\mathbf{U}\mathbf{U}^T ||_2 $ where $\mathbf{U}$ is top $K$ eigenvectors based on $s_c(.)$ for robust features and all eigenvectors except top $K$ for non-robust features. The objective is solved using gradient descent. Figure 6 shows the feature visualizations for a few classes in CIFAR-10 using the PGD adversarially trained ResNet-18 model. While the non-robust features appear to be more noisy than the robust features visually, the non-robust features might also have some useful information that reflects the drop in clean performance. However, it is not possible to argue based on the visual interpretation of the features since the difference is primarily coming from the eigenspace of the feature covariance.
>
> 2. **Discussion:** our robust and informative features definitions are inspired from Ilays et al (2019) as stated. We included a discussion on the difference between the definitions (robust features and informative features) to Ilyas et al (2019) after Definition 3.1 in Page 4 and Definition 3.5 in Page 6 in the updated version.
>
> ### Theoretical justification for the impact on the clean accuracy
> We note that the theoretical justification for the impact on clean accuracy is provided through Corollary 3.6 using the definition of informative features. The corollary shows that the top $K$ eigenvectors chosen based on the robustness score $s(u_i)$ are the most robust as well as the most informative. When $K=p$, we select the entire space that has the full information to give the best clean performance. Since $K<p$ in our experiments, it is bound to affect the clean performance negatively, however the experiments demonstrate that the percentage of drop in clean performance is marginal. We added this discussion in the updated draft. We also note that the explanation previously provided on this in Section 4.1 is detailed further.
> ### Adversarial attack based on RFI
> We thank the reviewer for this suggestion, and the preliminary exploration of the idea in the form of transfer attack evaluation is already presented in the initial submission. Table 2 presents the result for the attack generated from base model + RFI (right subtable) where we observe that RFI results in stronger adversary resulting in worse performance of the base model. In fact, for the C&W case, the base model completely loses its robustness. Given this observation, we expect RFI to be a stronger attack. However, exhaustive benchmarking against the existing adversarial attacks and a comparative study requires a different set of experiments, which deviates from our current focus on test-time defense. Hence, we leave the analysis of RFI based attack as future work.
>
> [Zeiler and Fergus, 2014] Visualizing and understanding convolutional networks. Computer Vision–ECCV 2014

---

### Review · Reviewer_DgBb · 2024-06-06

**Summary Of Contributions:**

Defense methods against adversarial attacks are important for developing reliable AI systems. However, most existing methods require more computational resources, huge datasets, or longer test times for inference. The authors provide a novel defense method that does not require more computational resources, datasets, or test time. They suggest the RFI method, motivated by the idea of robust features.

**Audience:**

Yes

**Broader Impact Concerns:**

It seems that there is no concern about ethical issues.

**Claims And Evidence:**

No

**Requested Changes:**

(Major) I strongly recommend that the authors test their method on EOT-combined attack methods. This would make their method more robust (see W1).

(Minor) I recommend that the authors show the experimental results on transformer-based models. If the proposed method does not work well, it would be better if the authors explain the reason for that.

**Strengths And Weaknesses:**

[S1] Their method is simple to adopt on any deep learning model and easy to combine with other defense methods. It could be highly effective in large-scale deep learning models.

[S2] They compare their method to various existing methods based on extensive experiments. Furthermore, it shows higher performance compared to other methods.

[W1] In [1], the authors suggest the expectation over transformation (EOT) method that can successfully attack models with any transformation of images. How robust is the RFI method against EOT-combined attack methods? Although the proposed method does not change the input, the EOT attack method can be modified for the RFI method.

[W2] Is the proposed method effective for transformer-based classification models? The motivation of this work, the idea of robust features, seems to target CNN-based classification models, so it may not work well on transformer-based models. However, it would be very beneficial if the authors show that this method works well on various models.

[1] Synthesizing Robust Adversarial Examples, ICML (2018)

---

> ### Author Response · Authors · 2024-06-16
>
> We thank the reviewer for the valuable feedback and comments. We provide results for the requested major experiment below.
>
> ### Performance of RFI with EOT combined attack models
> We evaluated ResNet-18 using all the robust training settings such as PGD, IAT, C&W and TRADES on CIFAR-10 for Expectation Over Transformation (EOT) as an adaptive attack. RFI can be easily integrated into the considered setups where the transformation matrix $\tilde{\mathbf{U}}$ is computed by applying random transformations to the training samples. The average performance with and without RFI is provided in the following table and detailed discussion is presented in Section B.3 and Table 9 in the updated draft. The results show that *RFI consistently improves the performance under EOT attack as well*.
>
> | Training | Clean      |        | $\ell_\infty(\epsilon = \frac{8}{255})$ |            | $\ell_2(\epsilon = 0.5)$ |           |
> |----------|------------|--------|-----------------------------------------|------------|--------------------------|-----------|
> |          | Method     | +RFI   | Method                                  | +RFI       | Method                   | +RFI      |
> | PGD      | **81.08** | 80.49  | 36.01                                   | **37.85** | 35.52                    | **36.65** |
> | IAT      | **90.32** | 89.89  | 26.92                                   | **28.30** | 30.30                    | **31.47** |
> | C&W      | **77.55** | 77.50  | 22.51                                   | **23.88** | 25.71                    | **26.95** |
> | TRADES   | **79.17** | 79.02  | 47.20                                   | **47.98** | 48.55                    | **49.61** |
>
>
> We thank the reviewer for the minor suggestion to include the experimental results on transformer-based models. While our work has primarily demonstrated the merit of RFI on convolution-based models such as WideResNets and PreActResNet, we agree that it would be interesting to consider transformer-based models as well. We are considering adding this model class to our future research.

---

### Decision · Action_Editor_eegM · 2024-07-27

**Recommendation:** Accept as is

**Comment:**

The paper is recommended for acceptance due to its strong theoretical basis, practical RFI method, and extensive empirical validation demonstrating improved adversarial robustness. Reviewers found the research interesting and well-presented, though noted a limitation in scalability, as the method is primarily evaluated on CNN-based models. It is suggested that the authors discuss this limitation in the camera ready version. The findings are relevant and valuable to the TMLR audience interested in adversarial robustness in AI.

**Audience:**

The topic of adversarial defense in deep learning is of significant importance to the machine learning community, particularly for those working on developing robust AI systems. The proposed RFI method offers a novel and practical approach to enhancing model robustness without additional computational overhead, making it relevant to researchers and practitioners focused on improving the security and reliability of AI systems. The comprehensive theoretical analysis and empirical validation further enhance the paper's appeal to this audience.

**Claims And Evidence:**

The claims made in the submission are supported by accurate, convincing, and clear evidence. The paper provides a thorough theoretical analysis justifying the proposed RFI method, demonstrates its applicability and ease of integration with existing models, and presents extensive experimental results across multiple datasets that show improved adversarial robustness. The empirical evidence, including the comparative analysis with other methods, validates the effectiveness and efficiency of the proposed defense mechanism.